



# The Portuguese Large Wildfire Spread Database (PT-FireSprd)

Akli Benali[1], Nuno Guiomar[2], Hugo Gonçalves[3,4], Bernardo Mota[5], Fábio Silva[3,4], Paulo M. Fernandes[6], Carlos Mota[3,4], Alexandre Penha[4], João Santos[3,4], José M.C. Pereira[1,7], Ana C.L. Sá[1]

[1]Centro de Estudos Florestais, Instituto Superior de Agronomia, Universidade de Lisboa, Tapada da Ajuda, 1349-017 Lisboa, Portugal
[2]MED - Mediterranean Institute for Agriculture, Environment and Development; CHANGE - Global Change and Sustainability; EaRSLab - Earth Remote Sensing Laboratory Institute; IIFA - Institute for Advanced Studies and Research; University of Évora, 7006-554 Évora, Portugal
[3]Força Especial de Proteção Civil, 2080-221 Almeirim, Portugal
[4]Autoridade Nacional de Emergência e Proteção Civil, 2799-51 Carnaxide,Portugal
[5]National Physical Laboratory (NPL), Climate Earth Observation (CEO), Hampton Rd. Teddington, TW11 0LW, UK
[6]CITAB - Centro de Investigação e de Tecnologias Agro-Ambientais e Biológicas, Universidade de Trás-os-Montes e Alto Douro, 5001-801 Vila Real, Portugal;
[7]Laboratório Associado TERRA, Tapada da Ajuda, 1349-017 Lisboa, Portugal

*Correspondence to*: Akli Benali (aklibenali@gmail.com)

**Abstract.** Wildfire behaviour depends on complex interactions between fuels, topography and weather, over a wide range of scales, being important for fire research and management applications. To allow for a significant progress towards better fire management, the operational and research communities require detailed open data on observed wildfire behaviour. Here, we present the Portuguese Large Wildfire Spread Database (PT-FireSprd) that includes the reconstruction of the spread of 80 large wildfires that occurred in Portugal between 2015 and 2021. It includes a detailed set of fire behaviour descriptors, such as rate-of-spread (ROS), fire growth rate (FGR), and fire radiative energy (FRE). The wildfires were reconstructed by converging evidence from complementary data sources, such as satellite imagery/products, airborne and ground data collected by fire personnel, official fire data and information in external reports. We then implemented a digraph-based algorithm to estimate the fire behaviour descriptors and combined it with MSG-SEVIRI fire radiative power estimates. A total of 1197 observations of ROS and FGR were estimated along with 609 FRE estimates. The extreme fires of 2017 were responsible for the maximum observed values of ROS (8956 m/h) and FGR (4436 ha/h). Combining both descriptors, we defined 6 fire behaviour classes that can be easily communicated to both research and management communities and support a wide number of applications. Analysis also showed that the area burned by a wildfire is mostly determined by its FGR rather than by its forward speed. Finally, we explored a practical example to show the PT-FireSprd database can be used to study the dynamics of individual wildfires and build robust case studies for training and capacity building.

The PT-FireSprd is the first open access fire progression and behaviour database in Mediterranean Europe, dramatically expanding the extant information. Updating the PT-FireSprd database will require a continuous joint effort by researchers and fire personnel. PT-FireSprd data are publicly available through https://doi.org/10.5281/zenodo.7495506 (last access: 30th



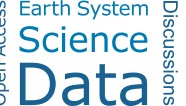

December 2022) and have a large potential to improve current knowledge on wildfire behaviour and support better decision-

making (Benali et al. 2022).

**Keywords:** fire behaviour; satellite; airborne; ground; rate of spread; fire radiative energy; graphs; progression



## 1 Introduction

Wildfire behaviour is broadly defined as the way a free-burning fire ignites, develops and spreads through the landscape (Albini 1984; Rothermel 1972). It depends on complex interactions between fuels, topography and weather, over a wide range of temporal and spatial scales (Santoni et al., 2011; Countryman, 1972). Wildfire behaviour can be described using common metrics such as the spread rate, propagation mode, area growth rate, perimeter, rate of energy release and flame size (Albini 1984). Fire behaviour information is important for fire research and management applications (Finney et al., 2021).

To allow for a significant progress towards better fire management, the operational and research communities require detailed open data on observed wildfire behaviour (Gollner et al., 2015). In this context, systematic mapping of the fire front progression through space and time is critical to address existing needs, for wildfires burning under a wide range of environmental conditions, including extreme ones (Storey et al., 2021; Gollner et al., 2015). Compiling quality fire behaviour information is paramount to develop reliable and well-suited fire spread models and for a much-needed extensive evaluation of fire behaviour predictions, which is crucial for its ultimate aim: support the decision-making process (Alexander and Cruz, 2013a; Scott and Reinhardt, 2001). This includes planning pre-suppression activities and defining resources dispatch to wildfires, delineating safe and effective fire suppression strategies and tactics during a wildfire, and for early alert and evacuation purposes (Finney et al., 2021). Comprehensive fire progression and behaviour information is also useful to develop burned area/fire perimeter mapping algorithms (Valero et al., 2018), understand fire effects (Collins et al., 2009), fire danger rating (Parisien et al., 2011), fire hazard mapping and risk analysis (Alcasena et al., 2021, Palaiologou et al., 2020), planning and implementation of preventive fuel treatments (Salis et al., 2018), and also to foster robust training of operative personnel and researchers improving their learnings from past wildfires (Alexander and Thomas, 2003). Unfortunately, reliable quality information on the progression and behaviour of wildfires, especially those burning under extreme conditions, is difficult to collect (Gollner et al., 2015).

Fire behaviour data can be collected from laboratory experiments, experimental fires, prescribed fires or wildfires. A large number of laboratory-scale experiments have been made for the development of semi-empirical rate-of-spread (ROS) models (Rothermel 1972; Catchpole et al., 1998). Experimental fires have been set up to collect fireline data, estimate fire behaviour descriptors and develop empirical fire spread models (Forestry Canada Fire Danger Group 1992; Fernandes et al., 2009; Cruz et al., 2015; Gollner et al., 2015), requiring significant time and resources. Neither laboratory-scale nor experimental fires represent the spatial and temporal variability of environmental conditions under which uncontrolled wildfires most often burn (e.g. Gollner et al., 2015).

Due to the unpredictability of their timing and location, conventional measurements on wildfires are difficult to perform and lead to slow accumulation of data (Alexander & Cruz 2013b). Generally, they are of poor quality or incomplete (Duff et al.,



2013), although outstanding reconstruction examples exist (e.g. Wade & Ward 1973; Alexander & Lanoville 1987; Cheney
2010). Dedicated efforts do exist (Vaillant et al., 2014), but wildfire behaviour estimates often result from opportunistic
observations (e.g. Santoni et al., 2011) or post-fire interviews (e.g. Butler and Reynolds, 1997). Some authors have made
relevant efforts in compiling a large amount of direct field observations on wildfire behaviour (Alexander and Cruz, 2006;
Cheney et al., 2012), some combined with experimental fire data (Cruz and Alexander, 2013, 2019; Anderson et al., 2015;
Cruz et al., 2018, 2021, 2022; Khanmohammadi et al., 2022). An additional limitation lies on the fact that some of the existing
fire behaviour datasets are not freely available for the operational and research communities (Gollner et al., 2015).

Remote sensing technology, either through airborne or satellite platforms, can provide relevant data to document wildfires.
Manned or unmanned airborne visible and infrared (IR) images have been collected to document fire progression, and in some
cases to retrieve fire radiative power estimates (Schag et al., 2021; Storey et al., 2020, 2021; Coen & Riggan 2014; Sharples
et al., 2012). Satellite data provide easy-to-use, autonomous, synoptic observations of fire activity throughout the entire globe.
Recent advances in satellite technology have made available a panoply of imagery and products that range from moderate to
high spatial resolution, and from every 5 days to sub-daily frequency. Several authors have used satellite data to map daily fire
progression at country-level (Parks et al., 2014; Veraverbeke et al., 2014, Briones-Herrera et al., 2020; Sá et al., 2017) and at
the global scale (Artés et al., 2019; Oom et al., 2016). Some have estimated fire behaviour metrics, such as ROS (Humber et
al., 2022; Frantz et al., 2017; Andela et al., 2019). Recently, Chen et al., (2022) improved this line of research by using Visible
Infrared Imaging Radiometer Suite (VIIRS) data to automatically reconstruct sub-daily fire progression at a higher resolution.
Other authors exploited the capabilities of geostationary satellites to monitor wildfires and estimate fire behaviour descriptors
(Sifakis et al., 2011; Storey et al., 2021).

The different data sources used to characterise wildfire progression and behaviour have inherent limitations and potentialities.
Ground-collected data can be characterised by large uncertainties, particularly when taken by fire personnel whose focus is on
suppression and not on data collection (Alexander and Thomas, 2003). In addition, ground-collected data have poor synoptic
capability and provide a limited representation of fire behaviour variability. For example, distribution of ROS values for single
fire runs are seldom available (Cruz, 2010). Airborne data can provide wider coverage of the fire progression, however, have
limited temporal acquisition windows (e.g. USFS National Infrared Operations - NIROPS - provides data once per night) and
in some cases require manual digitization of fire perimeters (Veraverbeke et al., 2014).

The tradeoff between spatial and temporal resolution of satellite data, as well as the presence of clouds and thick smoke can
significantly limit their fire monitoring capability. In addition, the correct location of a wildfire cannot be determined inside a
burning pixel whose size varies with viewing geometry and sensor properties (Wolfe et al., 1998). Daily or sub-daily satellite-
derived fire progressions can also fail to reflect the influence of extreme conditions in fire behaviour due to the effect of
averaging over relatively long periods (Collins et al., 2009).




Considering that all data sources have limitations and provide information for very limited time frames, combining different
sources is key to capture the spread and behaviour variability of wildfires. The example provided in Figure 1 highlights the
potential of combining different data sources to overcome inherent acquisition gaps, particularly in the afternoon, when both
field and airborne data overcome the satellite gap, and during dawn, when ground-collected and satellite data complement each
other. Note that observation frequencies of ground and airborne data strongly depend on daily fire activity patterns.

**(Figure 1 near here)**

Systematic multi-source acquisition of wildfire data collection was recently done by Kilinc et al., (2012) and Storey et al.,
(2020, 2021) for Australia, by Crowley et al., (2019) for Canada (only satellite data) and by Fernandes et al., (2020) at the
global scale. The pursuit of this goal requires a monitoring framework and a concerted joint effort between research and
operational communities (Stocks et al., 2004; McCaw et al., 2012, Storey et al., 2020, 2021). Additional data on constantly
evolving wildfires, accompanied by robust replicable methods, is needed, namely in southern Europe where a substantial data
gap is manifest (Fernandes et al., 2018).

Here, we present the Portuguese Large Wildfire Spread Database (PT-FireSprd) that combines data from multiple sources,
using a "convergence of evidence" approach to characterise in detail the progression and behaviour of large wildfires in
Portugal. Fire behaviour is described in *sensu stricto*, thus analysis of its drivers and effects is beyond the scope of the current
work. The work results from a joint co-creation effort between researchers and fire personnel, integrating data collected from
airborne and ground operational resources.
**2 Data and Methods**
**2.1 Overview**
We first collected data for all the large wildfires (>100 ha) that occurred in mainland Portugal  between 2015 and 2021. These
large wildfires were responsible for almost 1 million hectares burned during this period, of which half in the extreme fire
season of 2017. About 90% of the total burned area resulted from the 760 larger wildfires.
Multi-source input data (L0, section 2.2) were collected and only wildfires with good quality and representative data were
kept. Fire progressions were reconstructed from the input data and fire behaviour metrics were estimated. The PT-FireSprd
database was then organised in three levels:
● L1: Wildfire Progression (section 2.3), representing the spatial and temporal evolution of the wildfire spread (i.e.
where and when).



- L2: Wildfire behaviour (section 2.4), including quantitative behaviour descriptors of how a wildfire burned, such as the rate-of-spread (ROS), fire growth rate (FGR), fire radiative energy (FRE), and FRE flux;
- L3: Simplified Wildfire behaviour (section 2.5), averaging fire behaviour over longer periods that represent relatively homogenous fire runs.

The data from the different levels is composed by a large set of maps that can be useful for several applications and target users. For example, L1 data can be used by fire analysts or researchers to evaluate suppression strategies and understand the fire spread drivers or to evaluate burned area/fire perimeter mapping algorithms. L2 data is useful, for example, to calibrate existing or build better fire spread models, while potential applications of L3 are improving fire danger rating, fire hazard mapping and risk analysis. The overall flow of the data and methods is described in Figure 2.

**(Figure 2 near here)**

**2.2 Input Data (L0)**

To reconstruct the wildfire progressions, we used data acquired by satellites, from airborne sources and in the field by fire personnel. Most of this data is currently integrated in a near-real time operational WEB-GIS fire monitoring platform (in Portuguese "FEB Monitorização", hereafter FEBMON) developed in 2018 by the Civil Protection Special Force (FEPC) and the Portuguese National Authority for Emergency and Civil Protection (ANEPC). The data were complemented with official fire data (e.g., ignition date and location) and information from external reports.

**2.2.1 Satellite data**

Satellite data was used to support the reconstruction of past wildfire spread. Currently, there are many sources of open-access satellite data with capabilities to monitor wildfires over the entire globe. Their characteristics vary in resolution, ranging from high (10-30 m) to low (4-5 km), and frequency of overpass, ranging from 5-15 days to every 15 minutes. To monitor wildfire progression, satellites provide imagery and products that identify the location where a fire is actively burning at the time of overpass ("thermal anomalies" or "active fire" products).

The Sentinel-2 Multispectral Instrument (MSI) and the Landsat 8\9 Operational Land Imager (OLI) provide images of the Earth's surface on average every 5 days when combined. Their spatial resolution ranges between 10 and 60 m depending on the spectral band. PROBA-V has a lower number of spectral bands (4) when compared with other satellites used and provides daily images at 300 m of spatial resolution and every 5 days with a 100 m spatial resolution. The VIIRS instrument aboard the NPP and NOAA-20 satellites, collects data on average twice per day with a resolution varying between 375 m to 750 m, depending on the spectral band. The Moderate-Resolution Imaging Spectroradiometer (MODIS) is an instrument onboard the TERRA and AQUA satellites with spatial resolutions ranging from 250 m to 1000 m, depending on the spectral bands,



providing on average four daily revisits when combined. Sentinel-3 satellites have onboard the Sea and Land Surface
Temperature Radiometer (SLSTR) and the Ocean and Land Color Instrument (OLCI), with spatial resolutions ranging between
500 and 1000 m for the former, and 300 m for the latter. Data is acquired twice per day on average, but the OLCI does not
retrieve nighttime data.
We used L2 satellite imagery from the above-mentioned sensors to create false colour composites that could highlight burned
areas (low NIR, high SWIR reflectance), active flaming areas (high SWIR and/or TIR reflectance) and unburned vegetation
(high NIR reflectance). The bands used in the false colour composites depend on spectral characteristics of each sensor. Typical
false colour composites contain bands 12-8A-4 of Sentinel-2, bands 7-2-1 for MODIS and bands 1-2-4 for PROBA-V. Most
imagery was downloaded from Sentinel EO Browser (https://apps.sentinel-hub.com/eo-browser/), Worldview
(https://worldview.earthdata.nasa.gov/) and VITO-EODATA (https://www.vito-eodata.be/PDF/) which allow easy and fast
access to historical L2 data.
To complement the satellite imagery, we used the thermal anomaly products of VIIRS (VNP14IMGML-C1, Schroeder et al.,
2014, 2017) and MODIS (MCD14ML-C6, Giglio et al., 2003, 2016), with 375 m and 1 km resolution at nadir, respectively.
Data is available at fuoco.geog.umd.edu and FIRMS (https://firms.modaps.eosdis.nasa.gov/). These products allow estimating
the approximate location and timing of an active wildfire, and also provide an estimate of the fire radiative power (FRP), a
proxy of the radiant energy released per time unit and proxy for fuel consumption and fireline intensity. In addition, coarse
resolution data (~4 km) from the Spinning Enhanced Visible and Infrared Imager (SEVIRI) sensor onboard the Meteosat
Second Generation (MSG) geostationary satellite, was used to characterise the temporal evolution of fire activity using FRP
estimates every 15' (Wooster et al., 2015). Data is available at https://landsaf.ipma.pt/en/products/fire-products/frpgrid/. The
FRP detections associated with each wildfire were identified using a spatial-temporal nearest distance algorithm. An empirical
threshold derived from the analysis of a selected number of wildfires was used to account for the satellite pixel geolocation
and temporal reporting uncertainties. For each wildfire, the Fire Radiative Energy (FRE), and associated uncertainties, were
estimated by integrating FRP detections over 30' periods and by assuming a constant rate of energy release (Eq. 1):
$FRE_i = 0.0009 \times (\sum_{k=1}^{2} FRP_k),$          (1)
where index i indicates the 30' bin, index k indicates the 15' FRP value in MW, and the 0.0009 factor converts the sum into
TJ.

### 2.2.2 Airborne data

Some aeroplanes and helicopters that operate during wildfires collect photos and videos. Data are collected during the initial
attack (i.e. up to 90 min after the alert) by the heli-brigades of the National Guard (GNR) using their mobile phones, and
occasionally, during extended attack. Aeroplanes, operated by FEPC\ANEPC since 2018, are equipped with a gimbal that



contains visible and thermal cameras, collecting photos and videos during extended attack. In addition, helicopters that
coordinate aerial suppression, also collect valuable information regarding fire progression. Both data sources collect data only
during daytime, with a very small number of exceptions, at relatively low altitudes.

These airborne data are systematically uploaded in real-time in FEBMON since 2018, providing high quality information
regarding the probable location of the fire start, active flaming zones, and specially wildfire progression. It is noteworthy to
mention that airborne footage is not synoptic, as different parts of the wildfire (e.g. left flank vs. right flank) are captured at
different moments. These, depending on the fire extent and operational priorities can be characterised by significant time lags.

### 210    2.2.3 Ground data

The FEBMON system is linked to user-friendly portable tools that allow collection of georeferenced ground data during
wildfires. These tools are typically installed in mobile phones and tablets and are used by fire personnel from several
organisations (e.g., fire fighters, forest service). Ground-collected data consists of three main types: i) photos and videos; ii)
points that identify active flaming combustion, inactive flaming or smouldering or locations requiring mop-up activities; iii)
polygons that delineate an area burned until the time of acquisition (i.e. fire progression).

Besides the data automatically linked to FEBMON, valuable ad-hoc information can be used to reconstruct wildfire spread,
such as additional photos and videos captured on the ground, and post-fire interviews. In sum, data collected by fire personnel
in the field provided valuable spatiotemporal information regarding wildfire spread, ignition and/or wildfire re-activation.

### 220    2.2.4 Official fire data

The Forest Service (in Portuguese, "Instituto da Conservação Natureza e das Florestas (ICNF)") provides a fire database with
the final burned area perimeters for the entire country derived from a combination of field work and satellite data
(https://geocatalogo.icnf.pt/). We found some errors in the final perimeters that were corrected manually with Sentinel-2 or
Landsat 8/9 post-fire false colour composites (see section 2.2.1). In addition, for a very limited number of very large multi-
day wildfires, we used burned area perimeters provided by the Copernicus Emergency Management Service
(https://emergency.copernicus.eu/mapping/). The Forest Service also provides information regarding the wildfire start
location, mostly based on post-fire investigation done by GNR personnel (SGIF, https://fogos.icnf.pt/sgif2010/). Ignition data
have several known issues (Pereira et al., 2011) the most relevant of which, for the purposes of the present study, is the accuracy
of its exact location.

ANEPC manages the Operation Decision Support System (SADO) that includes information, such as i) date/hour of the
wildfire alert; ii) ignition location provided by first responders; and iii) a time log that seldom contains useful contextual
information on wildfire location at a given date/hour.



### 2.2.4 Reports

We also used ignition and fire progression data published in reports on the dynamics of the very large wildfires of June 2017, including the Pedrogão Grande wildfire, and October 2017 (Guerreiro et al., 2017, 2018; Viegas et al., 2019). Regarding Guerreiro et al. (2017, 2018), the primary data sources used to reconstruct the fire progression were satellite imagery, active fire data and burned area perimeters provided by the Copernicus Emergency Management Service (see 2.2.1). Reports from ANEPC and the Portuguese Institute for the Sea and the Atmosphere (IPMA, showing the fire plume evolution), GNR and the Association for the Development and Industrial Aerodynamics (ADAI), were also used to identify fire arrival times and active firelines. Additionally, other data sources allowed to reconstruct wildfire spread, such as: the official wildfire time log (see 2.2.4) , interviews (fire personnel involved in suppression, local residents), field work to identify the forward fire spread direction based on scorched or charred foliage orientation, and other relevant data such as photos and videos. The fire spread isochrones were determined through spatial interpolation methods (spline and inverse distance weighting), on high density point clouds and experts' knowledge.

Viegas et al., (2019) reconstructed the extreme wildfires of October 2017 based on field work, interviews, photos/videos and information contained in the official wildfire time log. Since the fire progression data were not provided by the authors, here we used only very limited information regarding ignition location\time and general fire spread patterns, mostly to complement data provided by Guerreiro et al., (2017, 2018).

We chose to include these fire progressions in our database, because they represent the most extreme wildfires that occurred in mainland Portugal, under persistent cloud cover conditions that limited the acquisition of satellite data, and for that reason they constitute relevant case studies, which otherwise would not be represented.

### 2.3 Wildfire Progression (L1)

Wildfire progression characterises the spatial and temporal evolution of the area burned in a specific fire event. It also contains information regarding the ignition time and location, as well as, flaming zones that correspond to active areas during the wildfire. These include spot fires and reactivation/rekindling areas. In Portugal, a rekindle is a reactivation of the wildfire after its official conclusion and is considered a new incident. For simplicity, we will consider rekindles as reactivations throughout the rest of the manuscript.

To robustly reconstruct wildfire progression, we combined the maximum available data from the different sources mentioned above, with the aim of obtaining convergence of evidence. This allowed reducing the limitations and uncertainties of each individual data source and building higher confidence in the derived wildfire progression.



Combining all the available data , we manually delimited the extent and time of the ignition, fire progression and active flaming
zones of each wildfire. The reconstruction was always made chronologically, i.e. starting from ignition and ending with the
progression prior to wildfire containment. Sentinel-2 and Landsat 8/9 pre-fire images were used to identify areas burned shortly
before the wildfire, and post-fire images were used to correct each progression polygon. As an example, Figure 3 shows how
different data sources were combined to derive the spread of the Castro Marim (2021) wildfire. All wildfire progression items
(L1) were defined as polygons, each with a set of different attributes (explained below).

**(Figure 3 near here)**

Ignition was defined as an area, instead of a point, to account for uncertainties in its location and to have a common data
typology for the entire database, in this case, vector polygons. We used mostly official ignition data and initial attack airborne
photos to define its location. This was complemented with expert knowledge and information from fire personnel to better
define ignition location. For a small set of wildfires (mostly nighttime ignitions), we also used satellite imagery and active-fire
data to identify the ignition area. All ignitions were compared with later fire spread patterns and with the final burned area to
reduce errors and guarantee consistency (e.g. ignition was contained in the final burned area). Regarding ignition time, the
official time of alert was compared with high frequency MSG-SEVIRI FRP detections, to confirm the alert time or, in a very
few cases, to anticipate if energy was released before the official ignition time. In addition, MSG-SEVIRI FRP were also useful
to identify (or confirm) the timing of reactivation. A clear example is shown in Figure 3, where the significant release of energy
around 11:30, combined with ground data, allowed identifying the location and time of the reactivation zone.

Active flaming zones were mostly derived from ground and/or high spatial resolution satellite imagery. Alternatively, they
were defined based on visual interpretation of multiple moderate resolution satellite imagery and often combined with active
fire data (mostly VIIRS due to its spatial resolution). Inconclusive visual interpretations were discarded, as well as active zones
that did not lead to any relevant subsequent fire spread. The ignition zone and all active flaming zones were always contained
within the subsequent fire spread polygon.

Wildfire progression was represented by a series of consecutive polygons delineating the temporal evolution of the area burned
by the wildfire. The number of polygons depended on fire size and data availability. The progression polygons were built using
as many data sources as possible, complementing each other in both space and time (see Figure 1). As an example: a common
feature found in the data was a pronounced fire spread during daytime, followed by very limited nighttime progression. In
these cases, first, the nighttime fire progression was delineated using active fire data (mostly VIIRS) and complemented with
ground data, when available. Second, satellite and/or airborne imagery acquired during the following morning were used to
perform any necessary adjustments in the nighttime spread polygon(s). Satellite-derived FRE estimates based on SEVIRI/MSG
were also used to identify if any substantial fire activity occurred between VIIRS/MODIS nighttime overpass and daytime



imagery (satellite and/or airborne). We assumed that fire activity decreased significantly when the wildfire released less than 0.5 TJ per 30' period, and anticipated the date/hour of the fire spread polygon accordingly. In smaller wildfires (<500 ha) this threshold was set to 0.1 TJ. These thresholds were defined empirically (see Discussion section). The entire procedure reduced the uncertainties associated with the delineation of the nighttime spread polygons. It should be noted that the fire behaviour within the time span of each progression polygon was unknown and, therefore, was assumed to be free burning in a homogeneous way (Storey et al., 2021). When data were insufficient to determine when a given area burned, the spread polygon was flagged as "uncertain".

Ignitions/active flaming zones were linked to the resultant spread polygon(s), by assigning a numeric label to a field called "zp_link", providing an explicit connection between both, and allowing to track the source of a given burned progression polygon. When information was insufficient, for example, the start of the progression polygon was unknown, zp_link was defined as "0". After all ignition(s), fire progressions and active flaming zones were defined, each wildfire was divided into burning periods. We assumed that each burning period contained relatively homogeneous fire runs that:

       i) were ignited by the same set of ignitions or active flaming zones;

       ii) did not exhibit large fire spread direction shifts (less than 45° of variation);

       iii) were not impeded by barriers (e.g. previously burned area) and;

       iv) did not exhibit significant changes in fire behaviour (e.g. large ROS variation).

Regarding the latter criterion, for example daytime and nighttime runs were usually separated in different burning periods even if criteria (i)-(iii) were fulfilled. By definition, a new active flaming zone always marked the beginning of a new burning period; however, not all burning periods started with an ignition or active flaming zone, since this depended on data availability.

When direct evidence of fire spotting was available (i.e. exact location/timing of the spot fire(s), typically from ground and/or airborne data), if the fire front(s) rapidly (under 1 hour) coalesced with the original fire front, fire progression was merged into a single polygon. In the remaining cases, typically associated with medium distance spotting and/or slow burning fire fronts, the spotting location was defined as a new active flaming zone setting, defining a new burning period. When the exact location/timing of the spot fire was not available, evidence of spotting consisted of observations of non-contiguous burned areas that resulted from the same wildfire. These were typically separated by rivers, lakes and settlements. In these cases, due to lack of data, the polygons separated from the major fire run were defined with zp_link=0 if the distance was larger than 200 m. No fire behaviour descriptors were calculated for these burned areas.

The definition of the burning period was always dependent on data availability and, in some cases, was subjective. For the progressions derived using only satellite data, the length of the burning period was mostly determined by the timing of the





satellite overpass(es) and the FRE temporal evolution. For the progressions derived from more detailed data, the above-
mentioned criteria were easier to fulfil. In a few cases, uncertainties in fire progressions led to slightly overlapping periods.
An example is shown in the Results section and implications are addressed in the Discussion section.
After collecting input data for a large number of wildfires only those with at least one valid progression and a valid
ignition/active flaming zone were kept. We eliminated all suspicious cases where uncertainties were large, for example, due
to the presence of persistent smoke or clouds in the satellite images or absence of valid ground data. The L1 wildfire progression
database was defined by a set of polygons with attribute fields (details in section 3). The date/hour of each ignition(s), fire
spread and active flaming zones (if applicable) were approximated to the nearest 30' period.
Fire progression data from external reports were adapted to the rationale of the fire database described above. Findings from
different reports for the same wildfire were compared and satellite data was used to complement and improve the original fire
progressions.

## 2.4 Wildfire behaviour (L2)

The estimation of fire behaviour descriptors was supported by the use of spatial graphs. A graph is a mathematical structure
composed of nodes (N) and edges (E), which connect the nodes (Dale and Fortin, 2010). Based on the fire spread polygons
(L1) (Figure 4a), we built a spatial directed graph (or digraph) where each node refers to a spread polygon, and each edge
connects two spread polygons (i.e nodes), with a valid link (i.e. zp_link>0). These two nodes burned at different times, one
earlier ($t_i$) and the other later ($t_j$). The value of each edge was defined as the time elapsed between two nodes ($\Delta t_{ij}$) (Figure
4b). A node can have an inward edge (where fire is being transmitted from) and an outward edge (where fire is being
transmitted to).
First, the nodes were connected only if the associated fire progression polygons were contiguous, had the same zp_link value
and burned at different timings. Second, only the edges corresponding to the shortest elapsed time between two nodes were
kept. The digraph allowed to formally structure the connections between fire spread polygons enabling the calculation of fire
behaviour descriptors.
To allow a better understanding of the methods used, a brief explanation based on the Ourique (2019) wildfire is provided. In
Figure 4, the number of the polygons on the left matches the number of nodes on the right. After its start (1), the wildfire
spread fast  to the south and burned the area delimited by polygon 2 in about 120'. Fire behaviour changed after the head run,
and the left flank became the head and made a run to the southeast, burning the area represented by polygons 4, 5, 6 and 7, in
about 180'. This fire behaviour change observed at t=120' determined the definition of two burning periods: one corresponding



to the initial head run, the other corresponding to head run from the left flank. The digraph was built with 7 nodes and 6 edges
with values ranging between 30' and 120'.

**(Figure 4 near here)**

Based on the fire progression (L1) and the corresponding di-graph, we calculated the following set of fire behaviour descriptors
(L2): forward ROS (m/h), spread direction (° from North), FGR (ha/h), and FRE (TJ). The polygons referring to areas burned
shortly before the fire analysed were removed from L2.

ROS was calculated for each node (Nj) with a valid inward edge (Eij) connecting it to a prior node (Ni). By definition, the
forward ROS refers to the head of the fire and was calculated considering the longest distance line connecting two consecutive
fire progression polygons (i.e. nodes). representing the fastest spread (Storey et al., 2021). The ground distance (Dij) between
each pair of polygons was calculated as follows:

● All ground distances between the polygon vertices of Ni and Nj were calculated, using the European Digital Elevation
Model (EU-DEM v1.1, https://land.copernicus.eu/imagery-in-situ/eu-dem/eu-dem-v1.1) resampled to 50 m spatial
resolution;
● For each vertex of the Nj polygon, only the shortest distance was kept and the corresponding pair of vertices, from
Ni and Nj, were stored;
● Dij was defined as the maximum of all shortest distances between vertices.

The ROS was calculated by dividing the distance (Dij) by the time elapsed between the pair of polygons (Δtij) and expressed
in m/h. We divided the ROS calculation in two distinct measures:

● Partial ROS (hereafter, ROSp) calculated between two consecutive polygons;
● Mean ROS (hereafter, ROSi), calculated between the ignition (or active flaming front) and a given spread polygon.

The spread direction was calculated using trigonometric rules considering the two above-mentioned vertices between two
polygons. The spread direction was calculated both for ROSp and ROSi, where the difference lies only on the origin polygon.
FGR was calculated dividing the burned area by each polygon/node (Aj) by the time elapsed between polygons (Δtij) and was
expressed in ha/h. An example of the calculation of these fire behaviour descriptors is shown in Figure 5.

**(Figure 5 near here)**






In addition to the standard fire behaviour descriptors, we also estimated the FRE for each progression polygon. This procedure
raised additional challenges. First, MSG-SEVIRI is affected by clouds and smoke, which can hinder the estimation of FRE for
some periods of the wildfires, or for their entire duration. Second, due to the coarse resolution of MSG-SEVIRI it was not
possible to calculate the FRE for each polygon directly. To circumvent this, FRE was calculated for each 30' bin from ignition
until the date/hour of the last wildfire spread polygon. In parallel, we estimated the area burned in each spread polygon every
30', using its start/end dates and assuming a constant FGR. Then, for each 30' bin, the total FRE was divided by weighting its
value by the proportion of area burned in each spread polygon. Finally, for each spread polygon the 30' FRE estimates were
summed only if they covered more than 70% of its duration ($\Delta t_{ij}$), to ensure that the total FRE was representative.

We also estimated the FRE flux rate (GJ ha$^{-1}$ h$^{-1}$) for each spread polygon by dividing the estimated FRE by the corresponding
burned area extent and its duration ($\Delta t_{ij}$). As FRE is highly dependent on the extent burning at a given time window, the FRE
flux can provide estimates closer to "instantaneous" values required for other applications.

**2.5 Simplified Wildfire behaviour (L3)**

We calculated simplified metrics representing a mean fire behaviour across each burning period. This enables higher-level
analysis of the data, but at the cost of losing detail and making simplifications to the calculation of the fire behaviour metrics.

The simplified ROS corresponded to the ROSi estimated for the last spread polygon of a given burning period i.e. the average
ROS between the start and the end of each burning period. FGR was defined as the sum of the area burned in the period divided
by its duration. The total FRE was calculated considering all energy released by the polygons burning within the burning
period, if FRE estimates covered more than 70% of the area burned.

**2.6 Quality Control and Quality Assurance (QC/QA)**

All L1 to L2, and L2 to L3 processing was done using Matlab scripts complemented with quality controls checks to identify
errors in the original L1 data. These included simple checks to incorrect field names, incoherent data format (e.g., date/hour),
and consistency on the fire spread structure defined by the di-graphs, as for example: i) time elapsed between node was always
positive;and ii) every spread polygon with a positive zp_link was always associated with a predecessor valid node (either of
"z" or "p" type), among others.

During the processing of L1 data to L2, we did frequent quality checks to identify potential errors, for example, null values of
ROS or FGR associated with valid fire spread polygons, fire progression polygons that did not have a known start/end date, or
did not have a known link to a preceding fire source (e.g., active flaming zone). In addition, we selected some wildfires and



made independent calculations of the ROS and FGR and compared them with the ones estimated using the developed Matlab
code. All these quality control steps assured that the data produced were reliable and of the best possible quality. The process
was iterative, requiring frequent corrections to the L1 data and the re-run of the quality check.

Finally, for each wildfire we defined a confidence flag that provides an overall information of how reliable the fire progression
data were. Although directly related to L1, ultimately it should also provide the user an estimate of the confidence associated
with L2 and L3. This was defined empirically based on the uncertainties that arose in the process of building the fire progression
polygons and was graded into a 5-level system where 1 refers to the lower quality and 5 to the highest quality (Table A1).
**3 Results**
**3.1 Overview of the PT-FireSprd database**
The PT-FireSprd database contains data for 80 large wildfires that occurred between 2015 and 2021. The individual wildfire
burned area extent ranges from 250 to 45,339 ha, with a mean and median area of 5,990 and 1,665 ha, respectively. The 80
wildfires were distributed throughout mainland Portugal, covering a wide range of environmental conditions (Figure 6). The
database spans a wide fire behaviour variability both between (e.g. Figure 6A,B,F) as well as within each wildfire (e.g. Figure
6C,E,D). The total burned area extent of the wildfires contained in the database was around 460,000 ha, which represents about
half of the area burned in the 2015-2021 period. On average, progression was reconstructed for 93% of the area burned by the
80 wildfires, leaving 7% deemed "uncertain". Wildfire behaviour descriptors were estimated for 88% of the burned area extent
(ca. 400,000 ha). The time elapsed between two consecutive fire progression polygons ranged between 30' and 14h30 with an
average value of 3h15. The mean duration of the burning periods was around 8h00, with a standard deviation of 4h50.

**(Figure 6 near here)**

A total of 1197 polygons with ROS and FGR estimates (L2) were derived from the progression data. We excluded very small
polygons (<25 ha) from further analysis, resulting in a dataset with 874 observations. Of the 1197 polygons, only 609 had FRE
estimates. Regarding L3 data, ROS and FGR were calculated for 241 burning periods (L3) and total FRE was only estimated
for 162 burning periods.

Overall, confidence in the database was lower for the earlier years (2015-2016) because input data was mostly from satellites.
In 2017, the quality increased due to the integration of i) ground data and ii) data from external reports that analysed the
extreme wildfires of June and October. From 2018 onwards, the integration of the monitoring aeroplanes, the creation of the
FEBMON system and the rapid availability of all the data that flows through it, significantly improved confidence of the
derived fire progressions.




The estimated forward ROS displayed a long-tail distribution (Figure 7, in log-scale) with a median value of 341 m/h and
average ROS of 746 m/h, representing large variability (std = 1071 m/h, cv = 143%). About 20% of the ROS values were
larger than 1000 m/h and about 9% were larger than 2000 m/h. The maximum observed ROS was 8956 m/h in the Lousã
wildfire of October 2017. The FGR distribution was highly skewed towards low values, with median and average values of 40
ha/h and 191 ha/h, respectively (sd = 438 ha/h, cv = 228%). About 10% of the observations had FGR larger than 500 ha/h and
only about 5% were larger than 1000 ha/h. The maximum observed FGR was 4436 ha/h in the Pedrogão Grande wildfire of
June 2017.

**(Figure 7 near here)**

The ROS distributions of the L2 and L3 datasets were similar. The largest differences were located in the lower and upper
tails, where the L3 ROS tends to be smoother due to the averaging procedure done over a longer time span. The FGR
distributions for L2 and L3 were also very similar, probably because all the polygon areas within a burning period are summed,
and the value does not result from an average. Differences were larger for more complex wildfires, for example with "finger
runs" (e.g. areas resulting from rapid propagation in a different direction than the dominant fire front).

We compared the histograms of L2 ROS and FGR for three aggregated confidence levels. The distribution of ROS estimates
for wildfires with lower confidence was slightly skewed towards lower values, when compared with higher confidence
estimates (Figure B1). The ROS distributions peak at 200 m/h, 500 m/h and 800 m/h for very low/low, moderate and high/very
high confidence, respectively, showing a clear relation between confidence and estimated ROS. Regarding FGR, very high
values above 500 ha/h were prevalent in wildfires with high and very high confidence progressions (Figure B2). Results are
similar if data from external reports for the extreme wildfires from June and October of 2017 are not included.

Estimated ROS and FGR were compared and percentiles 25, 50, 75, 90 and 97.5 were calculated for each variable
independently (Figure 8). The percentile values were simplified to enable a clear communication of results, especially between
researchers and fire personnel. The percentiles were translated into empirical classes, ranging from "very low" to "extreme"
fire behaviour. In general, as ROS increases so does the FGR. However, the relationship between ROS and FGR depends on
the morphology of the fire perimeter: elongated fast-spreading wildfires had relatively higher ROS and lower FGR (e.g. Figure
6B, C) and more complex burned area perimeters had relatively lower ROS and higher FGR (e.g. a flank run with an extensive
active fireline; see Figure 6A and the last polygons of Figures 6E and 6F). The dispersion tends to increase with higher
ROS/FGR values suggesting a progressively larger dependence on the burned area extent/perimeter. Identification of factors
determining such relationships is beyond the scope of this work. Nevertheless, wildfires with "Extreme" behaviour had both
very high values of ROS and FGR.




**(Figure 8 near here)**

Burned area extent is a relevant fire behaviour descriptor for researchers and fire management personnel. Analysis suggests
that the area burned by a wildfire is mostly determined by its FGR (r=0.84) rather than by the speed of the forward spread
(r=0.62; Figure 9a,b). The (cor)relations were lower using L2 data. As expected, FRE is highly correlated with burned area
extent (r=0.85, Figure 9c), and consequently of FGR. Correlation between ROS and average rate of energy release (TJ\h) is
lower (r=0.30, Figure 9d), however, there is a general direct relation between both descriptors.

**(Figure 9 near here)**

**2.2 Case study: The Castro Marim 2021 wildfire**
Here, we describe in detail the progression and behaviour of a specific wildfire to show how the PT-FireSprd database can be
used, for example, to analyse case studies, something often done by researchers and fire analysts.

The Castro Marim wildfire burned 5950 ha on the 16th and 17th of August of 2021. Figure 10 shows its reconstructed
progression (a) and associated ROS (b). Ignition occurred at nighttime (01:00) and a single run occurred towards SE until
approximately 08:30, defined as the first burning period. The mean ROS was 618 m/h, ranging between 321 and 957 m/h
(Figure 10c). The estimated FGR for the burning period was 43 ha/h, ranging between 33 and 77 ha/h, and the total FRE was
13 TJ (Figure 10d).

**(Figure 10 near here)**

Fire progression halted for about 3h until the wildfire reactivated around 11h30. It spread southwards until the head stopped
in an agricultural area around 19h30. In this second burning period, fire behaviour was significantly different from the first.
The mean ROS was ca. 1500 m/h, reaching a maximum value of 3720m/h between 16:30 and 17:30. On average, the fire grew
at a rate of 455 ha/h, however, significant variability was observed with values reaching 1236 ha/h coinciding with the ROS
peak. Framing the fire behaviour descriptors with the empirical classes represented in Figure 8, the behaviour in the second
burning period was often framed in the "Very High" class, i.e. between percentiles 90 and 97.5. As a consequence of the
behaviour exacerbation, the wildfire released around 38 TJ, with peaks of about 9 and 12 TJ observed during the afternoon.
The energy flux rate was highest between 16:00 and 16:30, coinciding with an abrupt increase in ROS (Figure 10d).



After the fire head stopped, a secondary head run stopped around 23:00 in a previously burned area (burning period 3). In the
follow-up, two left flank runs were observed, one until 02:30 and the other one, resulting from a reactivation, until 06:00, with
decreasing ROS, FGR and FRE. A secondary peak in the energy flux rate was estimated around 0:00, associated with an
increase in ROS and FGR.

Finally, in the Castro Marim wildfire burning periods 3 and 4 overlapped in time. A progression polygon in the rear/right flank
was delimited by fire personnel at 02:30, however the prior contiguous progression was identified at 16:30, suggesting a very
low burning flank, opposite to the fast burning part of the wildfire southwards. This overlap had no effect on the average ROS,
and only a very slight effect on the estimated FGR and FRE. However, users must be aware that burning periods seldom
overlap (~4% registered in the entire dataset), which may have implications in posterior analysis.
**4 Discussion**
**4.1 The PT-FireSprd database**
The PT-FireSprd is the first open access fire progression and behaviour database in the entire Mediterranean Europe. The
progression of 80 large wildfires that occurred in Portugal between 2015-2021 is reconstructed and fire behaviour descriptors
such as ROS, FGR and FRE are estimated, dramatically expanding the extant information (Palheiro et al., 2006; Rodriguez y
Silva & Molina-Martínez 2012; Fernandes et al., 2016). Wildfire progression was derived by converging evidence from
multiple data sources, which provides added credibility to the database. Wide variability in fire behaviour is covered, tackling
an important limitation pointed out by Cruz (2010). The approach presented will be used to update the database in the following
years for Portugal, and can be replicated in other countries, depending on data availability.

The large number of fire behaviour observations, both at the polygon level (L2) and at the burning period level (L3), provide
enough information for a wide variety of potential applications. For example, it can be used to: i) improve current knowledge
on the drivers affecting the behaviour of large wildfires; ii) calibrate existing or new models which ultimately should help to
better predict fire behaviour and support efficient fire management strategies (Alexander and Cruz, 2013a); iii) support the
construction of case studies by fire analysts and contribute to better training of fire personnel (Alexander and Thomas, 2003);
iv) contribute to improve operational fire suppression strategies; v) better understand how fire behaviour is linked to its effects
(Collins et al., 2009), and v) improve fire danger rating (Wotton, 2009). In addition, the fire behaviour classes described in
Figure 8 can assist fire suppression operations, including resources dispatching and decisions to fight or flee, or offensive vs
defensive strategies.

For several reasons, it is easier to collect information for larger wildfires than for smaller ones. The wide range in fire size
present in the PT-FireSprd database suggests that it is representative of wildfires burning under a broad range of conditions.





However, smaller wildfires (between 100 and 500 ha) are slightly under-represented in the database creating a potential bias.
This can be particularly relevant if one considers the proportion of smaller wildfires that occur every year. Thus, fire behaviour
descriptors may also be biased towards larger values which may have an implication, for example, on the fire behaviour classes
defined in Figure 8. Note that for typical fuel loads, say 15-20 t ha-1 (Fernandes et al., 2016), the third class in Fig. 8 already
corresponds to fires very difficult to control directly (Hirsch and Martell 1996). Nevertheless, these classes should be
considered as a first exploratory approach with the aim of creating a simple and clear communication baseline between
researchers and fire personnel based on quantitative fire behaviour data. Ultimately, the database will allow framing the
behaviour of new wildfires according to historical patterns. Adding smaller wildfires to the PT-FireSprd database will certainly
help to better represent a wider range of fire behaviour.

Confidence in the wildfires of 2015-2016 was lower than for the most recent ones due to relevant advances in operational fire
monitoring resulting in better quality and higher quantity of fire data. Since 2018, the FEBMON system has improved and
grown, providing larger quantity and higher quality data, thus leading to more reliable and detailed fire progression
reconstructions. The distribution of the duration of the spread polygons between 2015 and 2021 (Figure B3) shows
heterogeneity of the database across time, but also the evolution introduced by the implementation of the FEBMON system.
Results suggest that estimates of ROS and FGR might be underpredicted in wildfires with lower confidence, most probably
due to the lack of data to thoroughly cover the afternoon, but especially the early night period (i.e. between VIIRS/MODIS
day and nighttime overpasses, Figure 1). This issue is further discussed in section 5.2. The user must take into account the
characteristics of the database and can choose to use the entire or part of the dataset based on the confidence flag or year of
the wildfire.

The PT-FireSprd database is flexible and open, allowing the users to subset the data based on their needs and requirements.
For example, users can decide to work with fire behaviour descriptors at the polygon level (L2) or at the burning period (L3),
or can create their own subset depending on their objectives. The dataset is heterogeneous which is reflected in two main
components: the duration of the spread polygons and the burning periods, and the confidence flag associated with each wildfire.

Regarding the duration, the average time elapsed between two progression polygons was 3h30 and 8h15 for the burning
periods. Durations were large in 2015 and 2016 (median values above 9h), decreased significantly in 2017 with the integration
of hourly isochrones from Guerreiro et al., (2017, 2018), and have had median durations below 2h since 2019 (Figure B3).
Gollner et al., (2015) argued that fire progression observations need to be made in real-time with a 10-metre spatial resolution
every 10' to meet the needs of fire behaviour forecasting. However, in operational context the current objective is to predict
fire behaviour time intervals larger or equal to 30' (Cruz and Alexander, 2013). Considering the average duration of the burning
periods, that represent a single fire run, the average time elapsed between progression observations represents a good
compromise and a clear advance in current data. Regardless, users can subset the database based on the duration of either the





progression polygons or the burning periods. L3 descriptors can be useful to provide more homogeneous and normalised fire
behaviour descriptors, dampening the effect of the large variability in L2 durations, allowing, for example, a better comparison
between wildfires.

Finally, preliminary results suggest that considering both ROS and FGR can improve understanding of wildfire dynamics. The
relation between both is related to perimeter morphology and extent, and future work is needed to better understand the
underlying factors. Most importantly, FGR was a better explanatory variable of burned area extent than ROS. The practical
consequence is that large burned areas can be generated by wildfires with a moderate forward ROS but with large FGR of the
entire perimeter, which in turn is highly influenced by spread duration and perimeter extent. This should have implications for
both the research and operational communities. FRE was estimated for a lower number of spread polygons and burning periods
when compared with ROS and FGR. This was most likely due to the impact of clouds and smoke on MSG detections and the
relatively conservative minimum number of observations threshold (75%). FRE and burned area extent were closely related,
however, relations between FRE and ROS were poor/moderate. One of the possible reasons may be related with the need to
consider the effect of the active perimeter extent when comparing both descriptors.

## 4.2 Limitations and future improvements

The generic limitations of the input data have been thoroughly described in Section 1. In particular for Portugal some
limitations of the data must be pointed out. Fire progression perimeters and fire points collected in the ground by fire personnel
have relevant spatio-temporal uncertainties. For example, there is often a lag between the date/hour a polygon is drawn in the
ground and the actual date/hour it burned completely. Another relevant issue is that of data acquisition / reporting errors done
by fire personnel, which may be reduced by improved training and experience. The number of users of the FEBMON system
has been growing in recent years and, with adequate training, it is expected that the quality and quantity of ground data will
increase in upcoming years. In fact, over 27,000 aerial and 2,500 ground photos were taken in the year 2022 which represents
a relevant increase compared to previous years.

Regarding airborne data, the discussion can be separated into two components. First, initial attack photos, which can be
extremely useful to draw initial fire progression and infer probable ignition areas, are not collected for every wildfire to which
a helicopter is dispatched, and sometimes are of poor quality. Additional training and increasing the awareness of fire personnel
for the relevance of the data they collect is necessary. Second, aeroplane data are acquired at relatively low altitude, precluding
a synoptic view of the wildfire. Time lags between data acquisition for different parts of the wildfire (e.g. left vs. right flanks)
may be large and introduce relevant spatio-temporal uncertainties in the delineation of the fire progression. In addition,
perimeters are drawn manually and depend on the training and experience of the fire expert. In upcoming years, the integration
of new airborne sensors, specially with multispectral capability, the ability to perform high-altitude scans and the use of
automatic perimeter delimitation procedures (e.g., Valero et al., 2018) should improve data quality and reduce the time lags of



airborne fire observations. With this new capacity, it will be possible to integrate deep learning processes in the data analysis,
increasing both the quantity and quality of the available fire data. This integration will also allow a well-organised structure in
data collection, management and analysis, improving decision-support systems. Finally, the use of UAVs during nighttime
(pioneered in 2022 in Portugal) will complement aeroplane/helicopter data during periods of low data availability.

Regarding official fire data, errors in the delineation of final burned area perimeters and in the ignition location, often located
outside of the fire perimeter, need to be corrected to increase the quality of the PT-FireSprd database. Regarding satellite data,
implementing (semi-) automatic algorithms to delimit fire perimeters (e.g., Chen et al., 2022) will increase the availability of
fire perimeters and reduce the uncertainties associated with manual perimeter delimitation. Improvements in the spatial
resolution geostationary satellites, such as the recently launched Meteosat Third Generation (MTG), will certainly improve
fire behaviour estimates, as already observed in HIMAWARI-8 and last generation GOES satellites.

Regarding methodological uncertainties, the major challenge was to assign the correct date/hour to a specific burned area. For
example, when raw data sources indicated that an area burned but active areas were absent or small, there were always
uncertainties as to when it actually burned completely, which could lead to a relevant ROS/growth rate underestimation. These
uncertainties were larger between dusk until VIIRS overpass(es) and between the later and dawn. One approach to reduce
these uncertainties was to use FRE data to monitor the daily cycle of fire activity and help to better define the start/end date of
a progression polygon. The method was empirical and future work is needed to better define the thresholds for setting the
ignition or reactivation times, as well as the end of a fire progression. Exploratory analysis done in a few wildfires of the PT-
FireSprd database suggest that FRE has a significant drop after the head of the fire stops, which may take several minutes/hours
until reaching the FRE thresholds used. This moment is commonly accompanied by a flank growth that burns slower and
releases lower amounts of energy. These fire dynamics probably explain why ROS was likely underestimated in low
confidence wildfires and why FGR was less affected by data confidence. Improvements can be achieved in the future, through
the use of more sophisticated methods (e.g. change point detection), more ground observations during the head to flank run
transition, and higher spatial resolution data from geostationary satellites. Part of these improvements can be used to partially
update the 2015-2021 wildfires of the PT-FireSprd database.

In terms of characterising uncertainties and its effects, future work should also adopt a metrological approach to propagate
uncertainties to the descriptors, providing useful information to users. By providing an uncertainty assessment, the PT-FireSprd
database would be on the pathway of Fiducial Reference Measurement (FRM) compliance.

The continuous update of the PT-FireSprd database will require a joint effort by researchers and fire personnel. The automation
of data collection procedures (discussed above), as well as dedicated training to fire personnel, are key factors to guarantee
both the quality as well as a sustainable update of the database. In the upcoming years, other fire behaviour descriptors could



be included such as type of spread (surface vs. crown fire), fireline intensity, flame size, spotting (including maximum distance)
and/or PyroCb occurrence. Finally, methods described in the current work can be, at least partially, applied to many other fire-
prone areas of the globe and contribute to the much-needed data on observed wildfire behaviour.
**5 Data Availability**
The dataset contains generic metadata file with relevant information for each wildfire (Table A2), such as the fire ID, official
incident ID (ANEPC, 13 digit number), fire name, municipality, civil parish, start date, duration (hours), extent (ha), among
others. The fire name was defined as Municipality_DDMMYYYY, where DD is day, MM month and YYYY the year.  In
case more than one wildfire occurred in the same municipality on the same day, we added an additional string at the end of the
fire name (e.g. "_2").

The dataset is then divided in 3 Levels, with three corresponding folders:
● Fire Spread (L1): Each year has a separate folder that contains one folder per wildfire labeled with the fire name. It
contains a polygon shapefile with the attributes listed in Table A3.
● Fire behaviour (L2): A single polygon shapefile that contains all wildfires and estimated fire behaviour metrics for
each individual fire spread polygon. The attributes are listed and explained in Table A4.
● Fire behaviour (L3): A single polygons shapefile that contains the simplified fire behaviour metrics calculated for
each burning period. The attributes are described in Table A5.


The generic metadata is connected to L1 data through the fire name field, and to L2 and L3 through the fire "ID" field.

The data are freely available at https://doi.org/10.5281/zenodo.7495506 (last access: 30th December 2022; Benali et al. 2022).
We intend to update the database annually with wildfires from the current fire season and implement continuous improvements
to the procedure. Also, if additional information from past wildfires becomes available, we will update the database either by
changing existing fire spread polygons or by adding new wildfires. Updates for future years depend on the availability of input
data and associated funding.
**6 Conclusions**
The Portuguese Large Wildfire Spread Database (PT-FireSprd) is the first open access fire progression and behaviour database
available within Mediterranean Europe. It includes the reconstruction of the progression of 80 large wildfires that occurred in
Portugal between 2015 and 2021, that was derived by converging evidence from multiple data sources, which provides added
credibility to the database. PT-FireSprd contains a very large number of key fire behaviour observations, such as ROS, FGR



and FRE. Based on the statistical distribution of ROS and FGR, we defined 6 broad fire behaviour classes that can be easily
communicated to both research and management communities and support a wide number of applications, including better fire
management strategies. The PT-FireSprd has a large potential to contribute to the development of better fire behaviour
prediction tools, improve our current knowledge on wildfire dynamics, foster better operational training and contribute to
better decision-making. The approach will be used to continuously update the database in the following years for Portugal and
can be replicated in other countries/regions, depending on data availability. Improvements in data quality and the
implementation of automated methods are key factors for the regular update of the PT-FireSprd database in the future.
**Appendix A: Supporting material for the Methods**
(Table A1, Table A2, Table A3, Table A4 and Table A5 near here)
**Appendix B: Supporting material for the Results**
(Figure B1, Figure B2 and Figure B3 near here)
**Author Contribution**
AB and FS designed the study. AB, NG, HG, CM, JS carried out data processing and delimited fire progressions. BM carried
out FRE data processing. AB assembled the database, performed data analysis and wrote the first version of the manuscript.
All authors contributed to the interpretation of the results and writing of the manuscript.
**Competing interests**
The authors declare that they have no conflict of interest.
**Acknowledgements**
We thank Florian Briquemont for initial data processing and progression delimitation; FEPC personnel that provided relevant
fire to reconstruct some of the wildfires: Pedro Machado, Eduardo Marques, Marco Pires, Marco Lucas, Miguel Martins,
Daniel Santana and Vítor Caramelo. We would also like to thank other fire personnel that provided relevant fire data: João
Pedro Costa (AFOCELCA), José Silva (AFOCELCA), António Louro (CM-Mação), Sónia Oliveira (CM-Mação), Rui Lopes
(CBV Peso da Régua), Amélia Freitas (CM Caminha), Rui Pedro Fernandes (CBV Valença), Carlos Gomes (CBV Boticas),
António Ribeiro (ANEPC), Mário Silvestre (ANEPC), Emanuel Oliveira, and Elisio Pereira (CBV Porto de Mós). Finally, we
would like to thank ANEPC and FEPC for providing full access to their fire data enabling the development of the entire work.




This research was supported by the Forest Research Centre, a research unit funded by Fundação para a Ciência e a Tecnologia
I.P. (FCT), Portugal (UIDB/00239/2020), project foRester (PCIF/SSI/0102/2017) and FIRE-MODSAT II (PTDC/ASP-
SIL/28771/2017) also funded by FCT.

Akli Benali was funded by FCT through a CEEC contract (CEECIND/03799/2018/CP1563/CT0003). Nuno Guiomar was
funded by the European Union through the European Regional Development Fund in the framework of the Interreg V-A Spain-
Portugal program (POCTEP) under the CILIFO (Ref. 0753_CILIFO_5_E) and FIREPOCTEP (Ref.
0756_FIREPOCTEP_6_E) projects and by National Funds through FCT under the Project UIDB/05183/2020. Paulo
Fernandes contributed in the framework of the FCT-funded project UIDB/04033/2020. Ana Sá was supported under the
framework of the contract-program nr. 1382 (DL 57/2016/CP1382/CT0003).

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

**Figures**

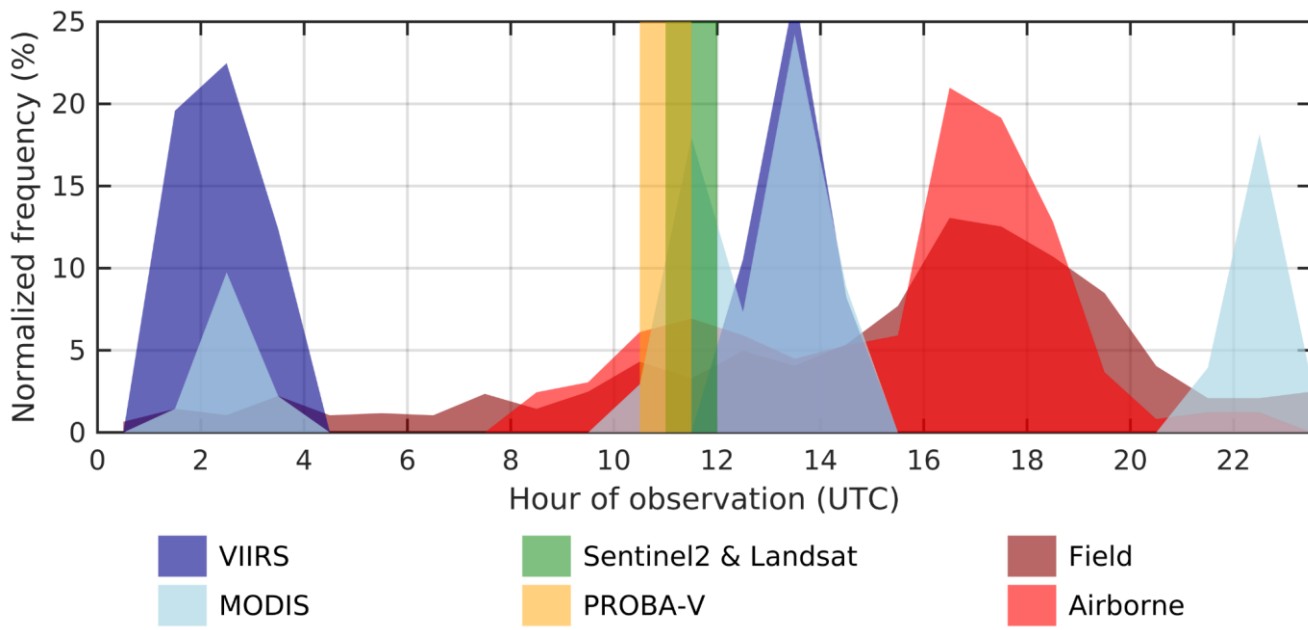

**Figure 1: Hourly frequency of observations in active wildfires acquisitions for satellite, field and airborne data. The data used refers to the year 2019 as an example. The frequency is normalised by dividing the number of observations by the total of each data source. Sentinel-2, Landsat and PROBA-V refer to the temporal windows and not the frequency, since all of the data are acquired in a very short window. The time windows of Sentinel-3 are similar to those of MODIS. MSG-SEVIRI data are not represented since it has a 15' frequency. Acronyms are described in the Data and Methods section.**





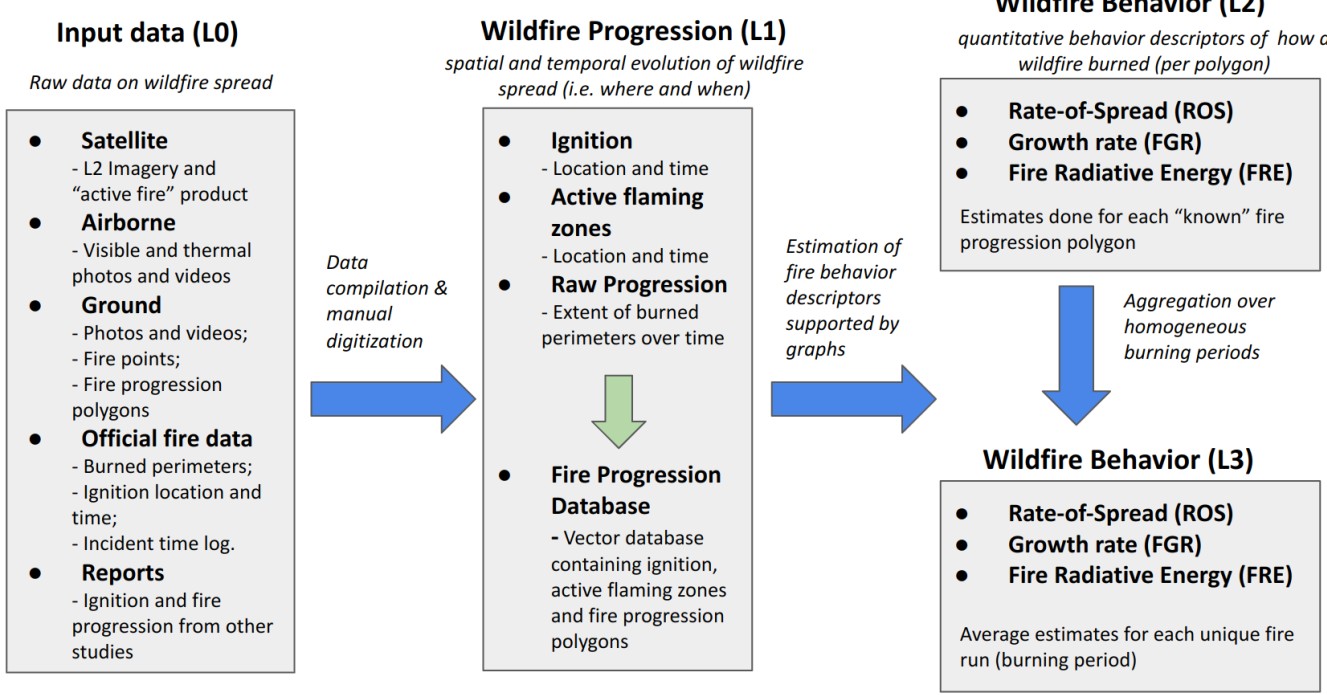

**Figure 2: Flowchart that represents an overview of the data and methods used in the development of the PT-FireSprd database.**

**Input data**

**Satellite data (VIIRS thermal anomalies)**

● 2021-08-16 03:09

● 2021-08-17 02:47

**Airborne and fire operatives data**

▼ A – Airplane, reativation (2021-08-16 11:32)

▼ B – Airplane, right flank (2021-08-16 16:26)

▼ C – Airplane (thermal), fire front (2021-08-16 16:38)

▼ D – Operatives, fire front (2021-08-16 19:30)

**Reports**

ⓘ Location reported in timeline (2021-08-16 16:18)

ⓘ Locations reported in timeline (2021-08-16 19:30)

**Estimated Fire Progression**

**Ignition/active flaming zones**

■ Reactivation Zone (2021-08-16 11:30)

**Fire perimeters**

■ 2021-08-16 03:00

■ 2021-08-16 09:00

■ 2021-08-16 16:30

■ 2021-08-16 19:30

■ 2021-08-17 03:00

■ 2021-08-17 12:00

□ Intermediate perimeters

**Fire Radiative Energy**



**Figure 3: Example of multi-source data integration to derive fire perimeters and reconstruct the progression of the Castro Marim**
**(2021) wildfire. The lines represent different progression polygons. Photos A, B, C, D were kindly provided by ANEPC\FEPC**





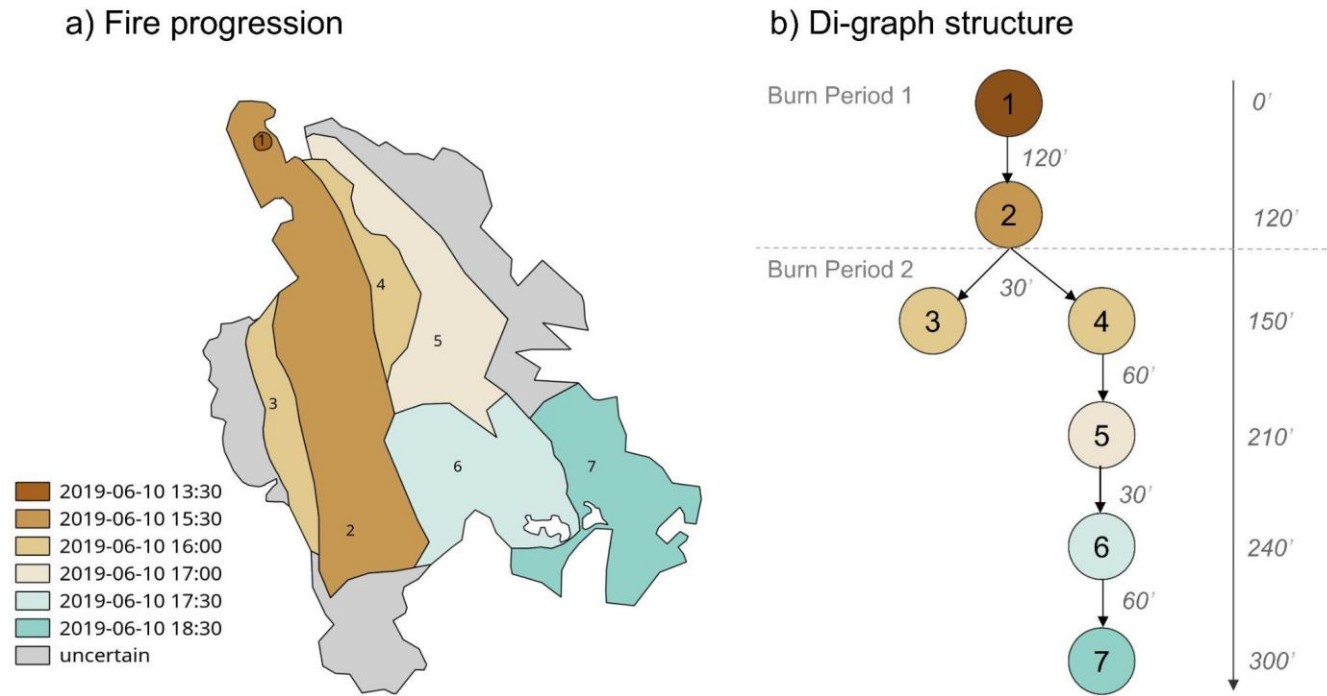

**Figure 4: Example of how the estimated fire progression (a) of the Ourique 2019 wildfire was used to build the digraph (b).**





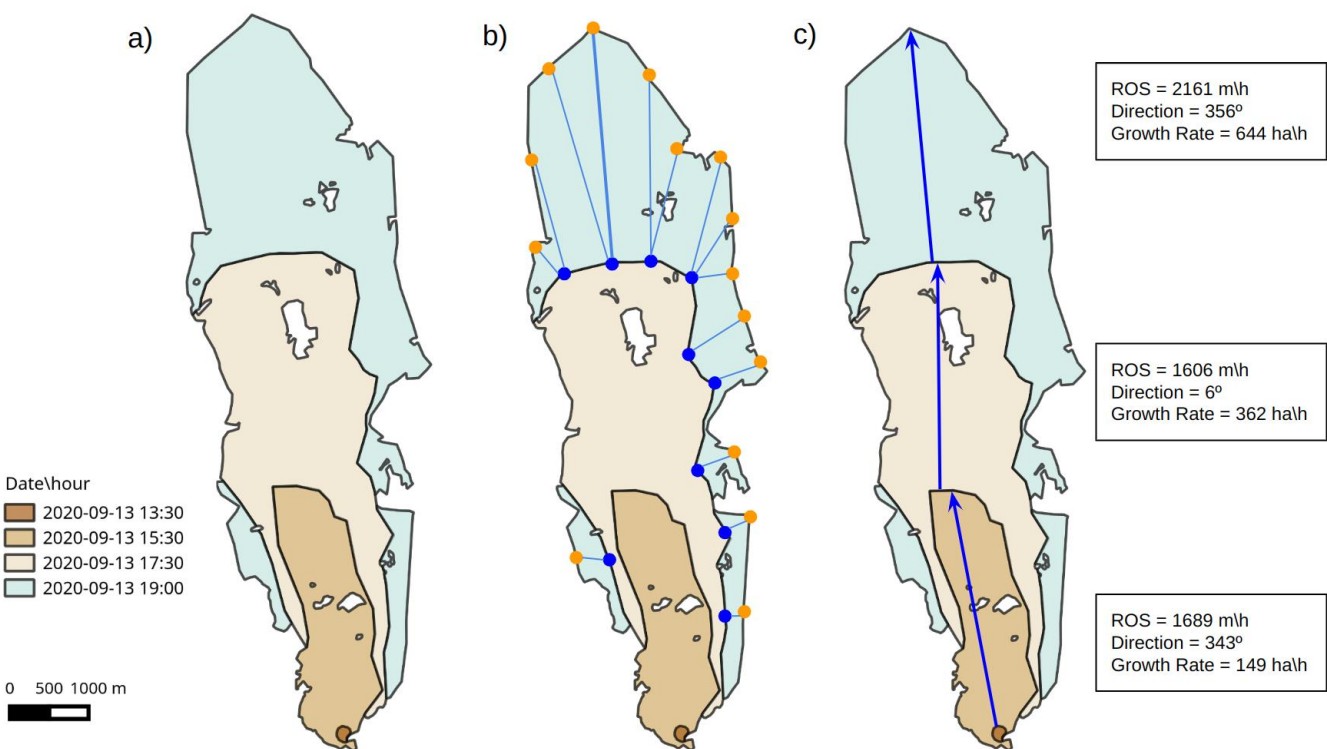



**Figure 5: Example of how the fire behaviour descriptors are calculated based on the Proença-a-Nova (2020) wildfire: a) partial fire**
**progression; b) procedure to calculate the distance for each vertex of the pair of consecutive polygons; and c) estimated main spread**
**axis and associated fire behaviour descriptors.**





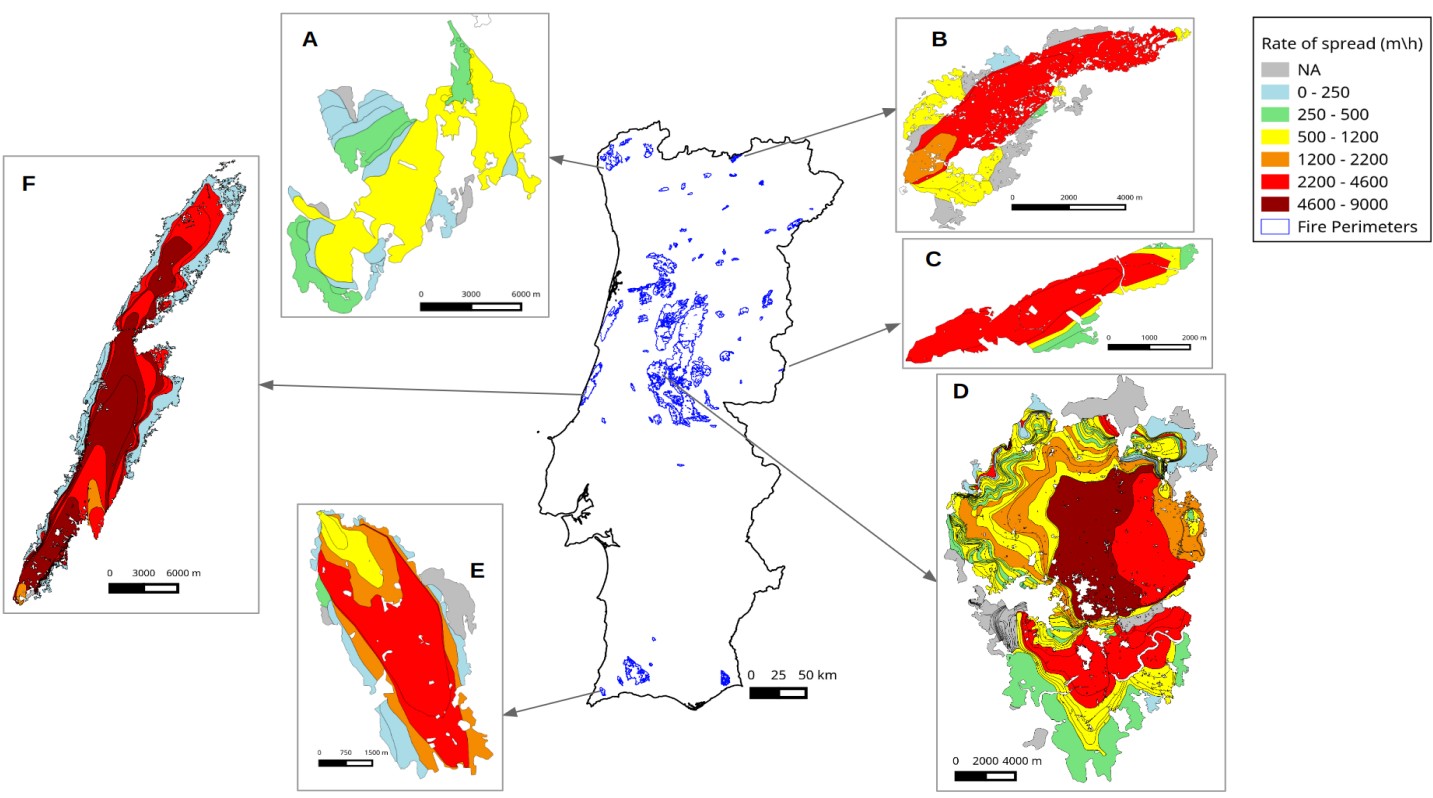

**Figure 6: Overall spatial distribution of the wildfire perimeters in the PT-FireSprd database, with examples of ROS estimates for 6 wildfires: A-Paredes de Coura (2016); B-Chaves (2020); C-Idanha-a-Nova (2020); D-Pedrógão Grande (2017); E-Aljezur (2020); F-Alcobaça (2017).**



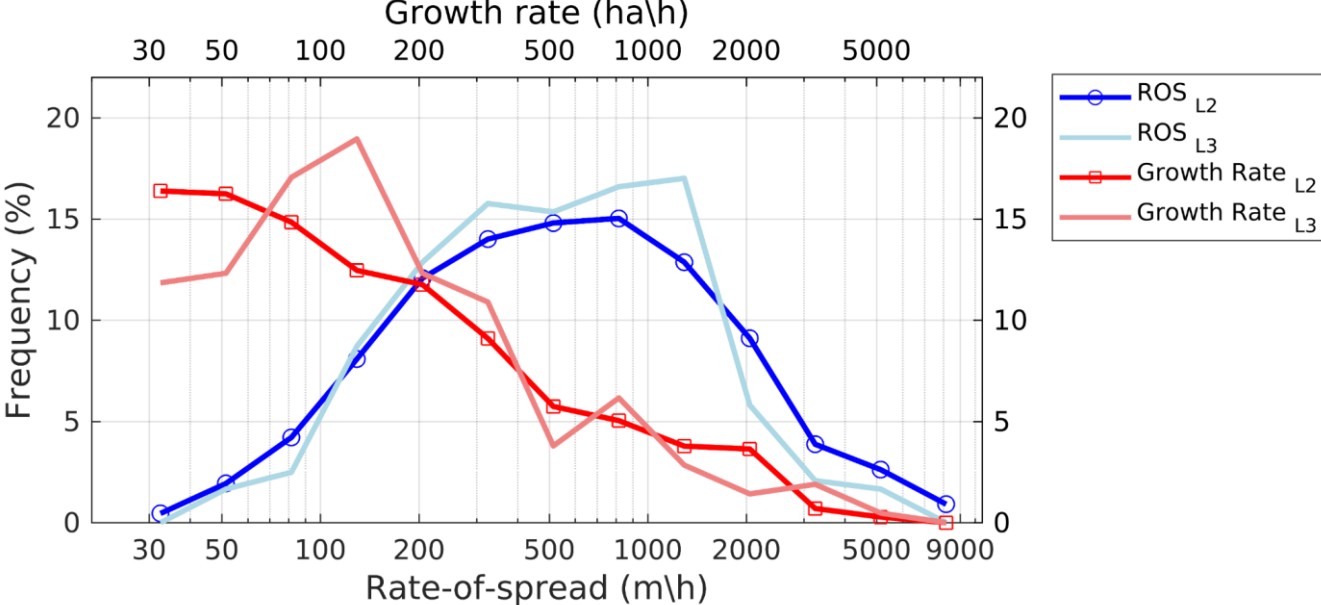


**Figure 7: Histogram of the estimated ROS and FGR for L2 and L3 data (in log-scale). Each point represents the frequency in evenly spaced bins on a logarithmic scale.**






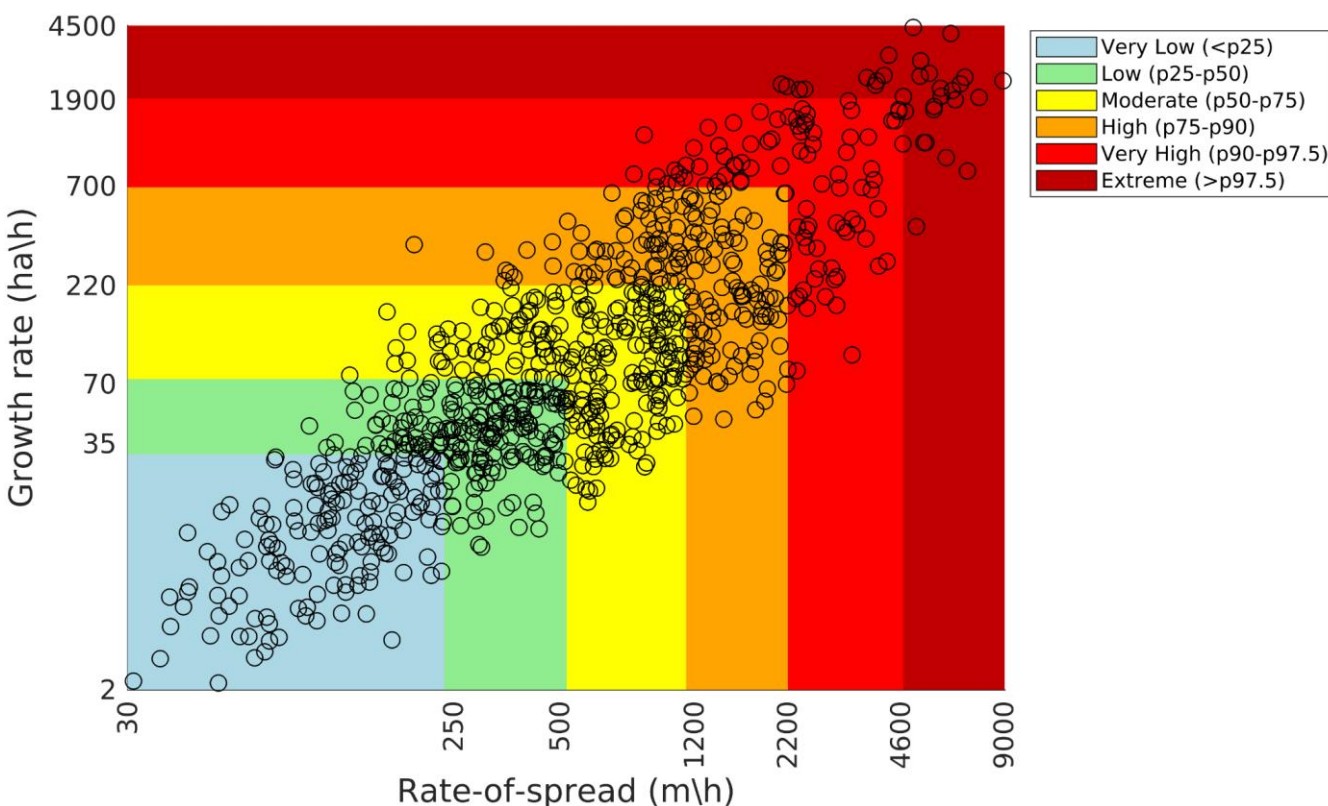


**Figure 8: Distribution of the estimated partial rate-of-spread (ROSp) and FGR (L2). Each point represents a wildfire progression with at least 25 ha of extent. The percentiles were calculated for each variable separately (n=874).**





**Figure 9: Comparison between simplified wildfire behaviour descriptors (L3): burned area extent and ROS (a), burned area extent and FGR (b), burned area extent and FRE (c), and ROS and average rate of energy release (d). The latter was calculated dividing the total FRE by the burning period duration.**


**Figure 10: The Castro Marim (2021) wildfire progression (a). Wildfire behaviour descriptors include: the spatial distribution of ROS (b); the temporal distribution of ROS and FRE flux rate (c); and the temporal distribution of FRE and FGR (d). Plots (c) and (d) start at 01:00 of the 16th of August and end at 06:00 of the 17th of August.**


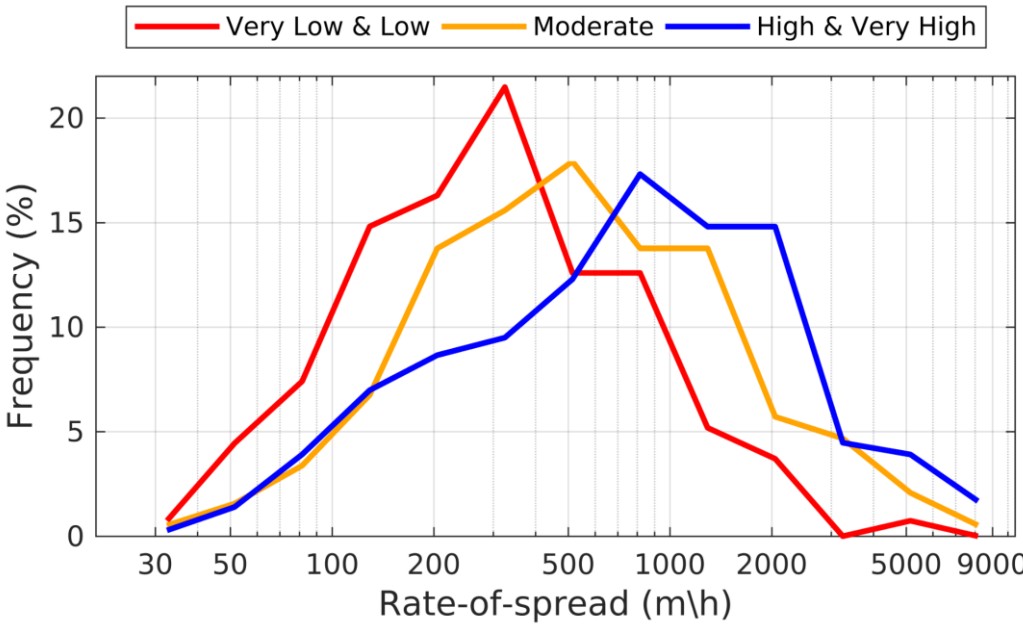


**Figure B1: Histogram of the estimated ROS (L2) for three aggregated levels of confidence. L2 ROS estimates were used and the confidence flags are explained in Table A1.**





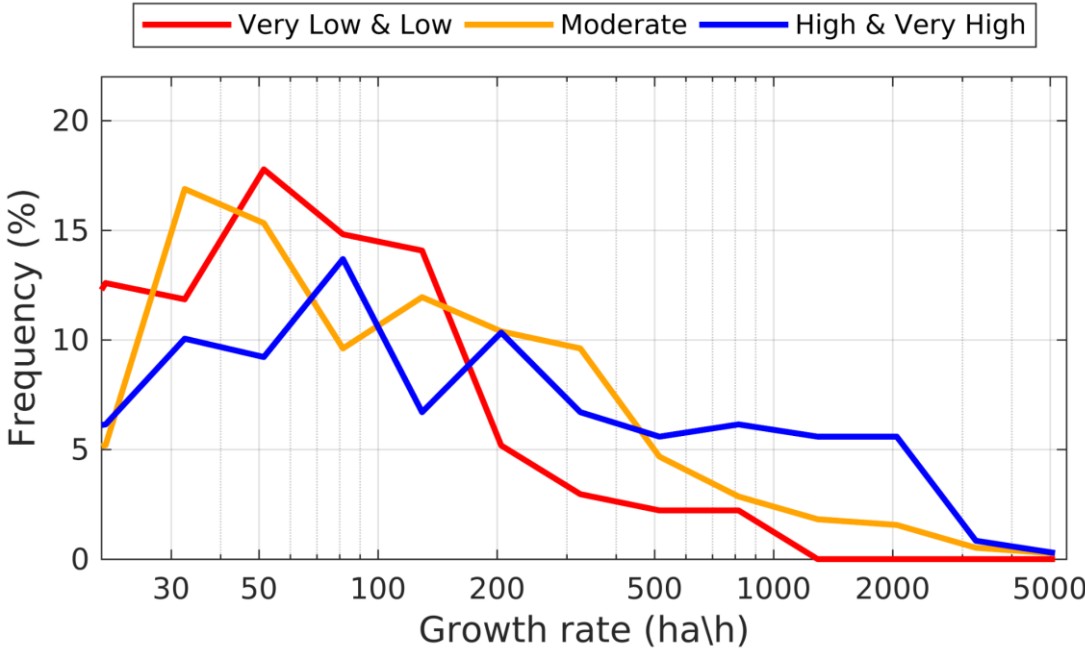


**Figure B2: Histogram of the estimated FGR for three levels of confidence. L2 FGR estimates were used and the confidence flags are explained in Table A1.**







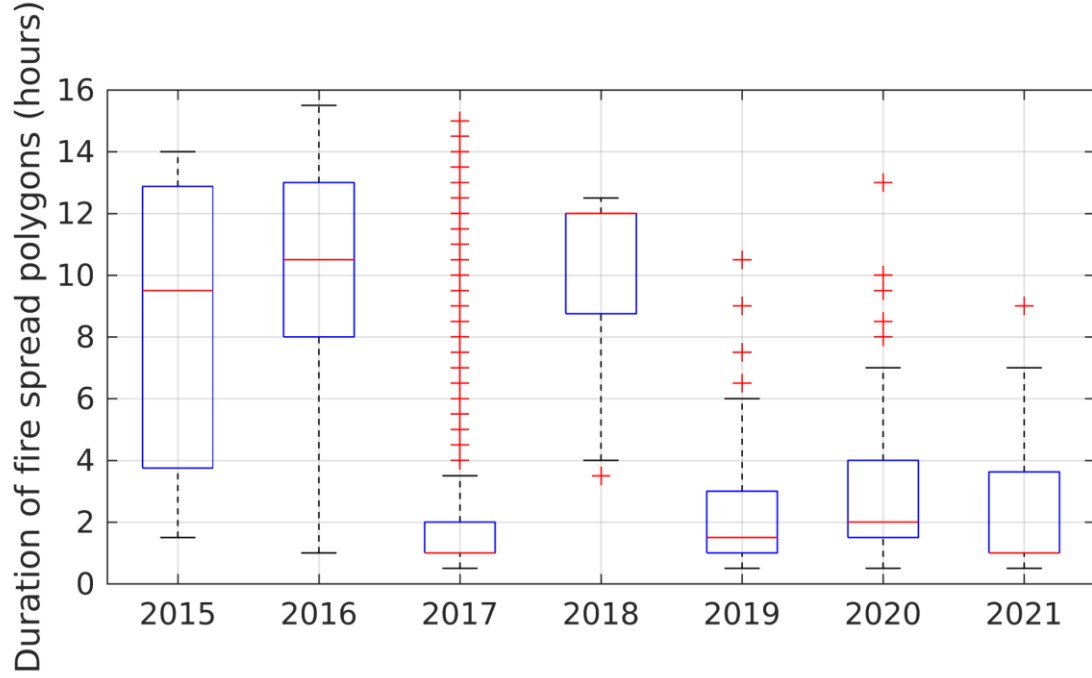


**Figure B3: Distribution of the duration of the progression polygons divided by years**



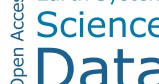

**Tables**
**Table A1. Confidence flag value, class and interpretation. The flag is defined for each wildfire.**

| Flag value | Flag Class | Interpretation |
|---|---|---|
| 1 | Very Low | The major fire progressions were observed only with satellite data, with important associated uncertainties. |
| 2 | Low | The major fire progressions were observed only with satellite data with moderate uncertainties |
| 3 | Moderate | The major fire progressions were observed with satellite data with low/moderate uncertainties and complemented with other sources. |
| 4 | High | The major fire progressions were at least partially observed with ground and airborne data, with relevant uncertainties associated (e.g. the exact hour of an important progression, or a flank position, etc) |
| 5 | Very High | The major fire progressions were observed with ground and airborne data with low uncertainties |






**Table A2. Database metadata list for L1**

| ID | Fire Name | Municipality | Civil Parish | Start Date | End Date | Extent (ha) | Confidence flag | ANEPC incident ID | P1 | P2 |
|---|---|---|---|---|---|---|---|---|---|---|
| 1 | Gouveia_10082015 | Gouveia | Mangualde da Serra | 2015-08-10 | 2015-08-12 | 2513 | 2 | 2015090024014 | 99 | 86 |
| 2 | Oleiros_03082015 | Oleiros | Alvaro | 2015-08-03 | 2015-08-04 | 853 | 2 | 2015050020535 | 100 | 95 |
| 3 | VilaNovadeCerveira_08082015 | Vila Nova de Cerveira | Candemil | 2015-08-08 | 2015-08-09 | 2988 | 3 | 2015160019994 | 87 | 87 |
| 4 | Agueda_08082016 | Águeda | Préstimo | 2016-08-08 | 2016-08-12 | 7317 | 1 | 2016010058351 | 99 | 63 |
| 5 | Anadia_10082016 | Anadia | V.N. de Monsarros | 2016-08-10 | 2016-08-12 | 3370 | 2 | 2016010059055 | 97 | 80 |
| 6 | ArcosdeValdevez_08082016 | Arcos de Valdevez | Cabana Maior | 2016-08-08 | 2016-08-11 | 5806 | 1 | 2016160022311 | 93 | 71 |
| 7 | Arouca_08082016 | Arouca | Janarde | 2016-08-08 | 2016-08-14 | 23547 | 2 | 2016010058554 | 97 | 96 |
| 8 | Boticas_05092016 | Boticas | Codecoso | 2016-09-05 | 2016-09-07 | 1694 | 3 | 2016170021732/ 2016170021835 | 97 | 97 |
| 9 | CabeceirasdeBasto_06092016 | Cabeceiras de Basto | Rio Douro | 2016-09-06 | 2016-09-07 | 1336 | 2 | 2016030067614 | 100 | 100 |
| 10 | Caminha_09082016 | Caminha | Argela | 2016-08-09 | 2016-08-11 | 1628 | 1 | 2016160022551 | 99 | 61 |
| 11 | Cinfaes_07082016 | Cinfães | Cinfães | 2016-08-07 | 2016-08-08 | 567 | 1 | 2016180042605 | 95 | 95 |
| 12 | Cinfaes_08082016 | Cinfães | Oliveira do Douro | 2016-08-08 | 2016-08-09 | 756 | 2 | 2016180042656 | 100 | 100 |
| 13 | FreixodeEspadaaCinta_06092016 | Freixo de Espada a Cinta | Freixo Espada à Cinta e Mazouco | 2016-09-06 | 2016-09-07 | 5194 | 3 | 2016040027372 | 99 | 97 |
| 14 | Moncao_06092016 | Monção | Riba de Mouro | 2016-09-06 | 2016-09-07 | 656 | 2 | 2016160025950 | 71 | 58 |
| 15 | Moncao_09082016 | Monção | Barroças e Taias | 2016-08-09 | 2016-08-11 | 1115 | 1 | 2016160022460 | 77 | 77 |
| 16 | ParedesdeCoura_07082016 | Paredes de Coura | Meixedo | 2016-08-07 | 2016-08-12 | 10457 | 2 | 2016160022456 | 100 | 96 |
| 17 | PontedeLima_08082016 | Ponte de Lima | Calheiros | 2016-08-08 | 2016-08-09 | 739 | 1 | 2016160022390 | 91 | 75 |
| 18 | SeverdoVouga_09082016 | Sever do Vouga | Pessegueiro do Vouga | 2016-08-10 | 2016-08-12 | 1818 | 3 | 2016010058973 | 96 | 94 |
| 19 | VieiradoMinho_10082016 | Vieira do Minho | Rossas | 2016-08-10 | 2016-08-11 | 1637 | 2 | 2016030060428 | 99 | 96 |
| 20 | Resende_17082017 | Resende | S. Martinho de Mouros | 2017-08-17 | 2017-08-21 | 544 | 1 | 2017180043566 | 84 | 38 |
| 21 | RibeiradePena_15082017 | Ribeira de Pena | Cerva | 2017-08-15 | 2017-08-16 | 507 | 1 | 2017170021591 | 100 | 100 |
| 22 | CastroDaire_05102017 | Castro Daire | Almofala | 2017-10-05 | 2017-10-05 | 701 | 2 | 2017180054022 | 99 | 99 |
| 23 | Mortagua_07102017 | Mortagua | Espinho | 2017-10-07 | 2017-10-08 | 961 | 2 | 2017180054507 | 99 | 99 |
| 24 | Mirandela_16072017 | Mirandela | Alvites | 2017-07-16 | 2017-07-17 | 949 | 2 | 2017040020105 | 100 | 88 |
| 25 | Pombal_06102017 | Pombal | Abiul | 2017-10-06 | 2017-10-07 | 1225 | 2 | 2017100054724 | 100 | 100 |
| 26 | TorredeMoncorvo_18072017 | Torre de Moncorvo | Acoreira | 2017-07-18 | 2017-07-18 | 1536 | 3 | 2017040020365 | 100 | 100 |
| 27 | Guarda_23082017 | Guarda | Fernão Joanes | 2017-08-23 | 2017-08-25 | 3457 | 3 | 2017090026098 | 91 | 91 |
| 28 | Serta_08092017 | Serta | Pedrogao Pequeno | 2017-09-08 | 2017-09-09 | 4177 | 3 | 2017050027511 | 100 | 100 |





| 29 | Abrantes_09082017 | Abrantes | Aldeia do Mato | 2017-08-09 | 2017-08-10 | 4357 | 3 | 2017140045924 | 83 | 79 |
|---|---|---|---|---|---|---|---|---|---|---|
| 30 | CasteloBranco_23072017 | Castelo Branco | Santo André das Tojeiras | 2017-07-23 | 2017-07-28 | 4569 | 3 | 2017050023219 | 97 | 85 |
| 31 | Serta_15102017_2 | Serta | Pedrógão Pequeno | 2017-10-15 | 2017-10-16 | 2320 | 3 | 2017050030728 | 54 | 54 |
| 32 | CasteloBranco_13082017 | Castelo Branco | Louriçal do Campo | 2017-08-13 | 2017-08-15 | 6173 | 2 | 2017050025136 | 100 | 96 |
| 33 | PampilhosadaSerra_06102017 | Pampilhosa da Serra | Fajao | 2017-10-06 | 2017-10-09 | 7217 | 2 | 2017060044928 | 97 | 96 |
| 34 | Guarda_17072017 | Guarda | Rochoso | 2017-07-17 | 2017-07-18 | 7523 | 2 | 2017090021641 | 88 | 88 |
| 35 | FigueiradaFoz_15102017 | Figueira da Foz | Quiaios | 2017-10-15 | 2017-10-17 | 15141 | 4 | 2017060046330 | 100 | 97 |
| 36 | Oleiros_23082017 | Oleiros | Cambas | 2017-08-23 | 2017-08-25 | 7985 | 3 | 2017050026111 | 88 | 67 |
| 37 | Gois_17062017 | Gois | Alvares | 2017-06-17 | 2017-06-22 | 15852 | 3 | 2017060026571 | 100 | 99 |
| 38 | Alcobaca_15102017 | Alcobaca | Pataias | 2017-10-15 | 2017-10-16 | 18575 | 4 | 2017100056537 /2017100056554 | 100 | 100 |
| 39 | Arganil_15102017 | Arganil | Coja | 2017-10-15 | 2017-10-16 | 31970 | 3 | 2017060046312 /2017090031521 | 100 | 99 |
| 40 | Serta_15102017 | Serta | Figueiredo | 2017-10-15 | 2017-10-17 | 30974 | 4 | 2017050030693 | 97 | 97 |
| 41 | Alvaiazere_11082017 | Alvaiazere | Pussos | 2017-08-11 | 2017-08-19 | 23715 | 2 | 2017100043917/ 2017050025201 | 99 | 52 |
| 42 | PedrogaoGrande_17062017 | Pedrogao Grande | Pedrogao Grande | 2017-06-17 | 2017-06-19 | 29456 | 4 | 2017100032538 | 92 | 91 |
| 43 | Serta_23072017 | Serta | Várzea dos Cavaleiros | 2017-07-23 | 2017-07-27 | 33401 | 3 | 2017050023195 | 97 | 96 |
| 44 | Lousa_15102017 | Lousã | Vilarinho | 2017-10-15 | 2017-10-17 | 45249 | 4 | 2017060046260 | 100 | 95 |
| 45 | Agueda_15102017 | Agueda | Albitelhe | 2017-10-15 | 2017-10-16 | 9095 | 3 | 2017180056272 | 83 | 78 |
| 46 | OliveiraFrades_15102017 | OliveiraFrades | Varzielas | 2017-10-15 | 2017-10-17 | 9297 | 3 | 2017180056290 | 99 | 97 |
| 47 | Monchique_03082018 | Monchique | Monchique | 2018-08-03 | 2018-08-08 | 26227 | 3 | 2018080033743 | 93 | 82 |
| 48 | Agueda_05092019 | Agueda | Macinhata do Vouga | 2019-09-05 | 2019-09-06 | 1602 | 3 | 2019010072794 | 89 | 84 |
| 49 | Alijo_24072019 | Alijo | Vila Verde | 2019-07-24 | 2019-07-24 | 574 | 5 | 2019170019467 | 100 | 100 |
| 50 | Baiao_04092019 | Baião | Teixeira | 2019-09-05 | 2019-09-06 | 728 | 3 | 2019130150620 | 75 | 73 |
| 51 | Nisa_01082019 | Nisa | Tolosa | 2019-08-01 | 2019-08-01 | 712 | 5 | 2019120016787 | 99 | 98 |
| 52 | Ourique_10062019 | Ourique | Monte Lavarjao | 2019-06-10 | 2019-06-10 | 554 | 5 | 2019020015472 | 75 | 75 |
| 53 | Penedono_21072019 | Penedono | Beselga | 2019-07-21 | 2019-07-23 | 736 | 4 | 2019180039496 | 99 | 99 |
| 54 | Sabugal_29082019 | Sabugal | Vale Mourisco | 2019-08-29 | 2019-08-29 | 578 | 5 | 2019090029579 | 100 | 100 |
| 55 | Serta_13092019 | Sertã | Marmeleiro | 2019-09-13 | 2019-09-14 | 676 | 4 | 2019050028005 | 100 | 90 |
| 56 | Tomar_03082019 | Tomar | São Pedro Tomar | 2019-08-03 | 2019-08-03 | 511 | 4 | 2019140045796 | 86 | 73 |
| 57 | Valenca_04092019 | Valença | Cerdal | 2019-09-04 | 2019-09-05 | 642 | 1 | 2019160026115 | 83 | 83 |
| 58 | Valpacos_13092019 | Valpaços | Ervões | 2019-09-13 | 2019-09-13 | 738 | 2 | 2019170026369 | 56 | 56 |
| 59 | ViladeRei_20072019 | Vila de Rei | Fundada | 2019-07-20 | 2019-07-22 | 9305 | 3 | 2019050022178 | 99 | 99 |
| 60 | MirandadoCorvo_13092019 | Miranda do Corvo | Moinhos | 2019-09-13 | 2019-09-14 | 540 | 3 | 2019060042282 | 96 | 96 |





| 61 | Fundao_07082020 | Fundão | Capinha | 2020-08-07 | 2020-08-08 | 472 | 4 | 2020050018968 | 87 | 85 |
|----|-----------------|--------|---------|------------|------------|-----|---|---------------|----|----|
| 62 | Silves_06072020 | Silves | Boião | 2020-07-06 | 2020-07-06 | 520 | 4 | 2020080025576 | 77 | 77 |
| 63 | Avis_21072020 | Avis | Montes Juntos | 2020-07-21 | 2020-07-21 | 698 | 5 | 2020120014122 | 95 | 95 |
| 64 | IdanhaaNova_30062020 | Idanha-a-Nova | Salvaterra do Extremo | 2020-06-30 | 2020-06-30 | 728 | 4 | 2020050015270 | 100 | 100 |
| 65 | SaoJoaoPesqueira_10072020 | São João da Pesqueira | Riodades | 2020-07-10 | 2020-07-11 | 770 | 4 | 2020180031783 | 97 | 94 |
| 66 | Fundao_06082020 | Fundao | Bogas Baixo | 2020-08-06 | 2020-08-06 | 749 | 5 | 2020050018872 | 96 | 96 |
| 67 | PortoMos_06092020 | Porto de Mós | Codacal | 2020-09-06 | 2020-09-07 | 998 | 4 | 2020100046280 | 97 | 91 |
| 68 | OliveiraFrades_07092020 | Oliveira de Frades | Antelas | 2020-09-07 | 2020-09-08 | 1902 | 3 | 2020180044235 | 86 | 73 |
| 69 | Aljezur_19062020 | Aljezur | Bordeira | 2020-06-19 | 2020-06-20 | 2243 | 5 | 2020080023014 | 99 | 93 |
| 70 | Sernancelhe_06082020 | Sernancelhe | Lapa | 2020-08-06 | 2020-08-06 | 2213 | 5 | 2020180037681 | 100 | 100 |
| 71 | Chaves_30072020 | Chaves | Vila Verde da Raia | 2020-07-30 | 2020-07-31 | 2508 | 3 | 2020170018342 | 83 | 82 |
| 72 | Oleiros_25072020 | Oleiros | Sardeiras de Baixo | 2020-07-25 | 2020-07-27 | 5564 | 3 | 2020050017687 | 95 | 92 |
| 73 | ProencaaNova_13092020 | Proenca-a-Nova | Cunqueiros | 2020-09-13 | 2020-09-14 | 14568 | 4 | 2020050022403 | 91 | 91 |
| 74 | CasteloBranco_29082020 | Castelo Branco | Ponsul | 2020-08-29 | 2020-08-29 | 315 | 4 | 2020050021105 | 100 | 92 |
| 75 | CastroDaire_07092020 | Castro Daire | Cujo | 2020-09-07 | 2020-09-07 | 452 | 4 | 2020180044155 | 76 | 76 |
| 76 | Odemira_18082021 | Odemira | João Martins | 2021-08-18 | 2021-08-19 | 944 | 5 | 2021020019189 | 100 | 98 |
| 77 | CastroMarim_16082021 | Castro Marim | Pernadeira | 2021-08-16 | 2021-08-17 | 5956 | 5 | 2021080035488 | 100 | 99 |
| 78 | Monchique_17072021 | Monchique | Tojeiro | 2021-07-17 | 2021-07-18 | 1900 | 4 | 2021080029244 | 99 | 99 |
| 79 | FreixoEspadaaCinta_20082021 | Freixo de Espada à Cinta | Lagoaça | 2021-08-20 | 2021-08-20 | 412 | 4 | 2021040023667 | 71 | 71 |
| 80 | Mogadouro_20072021 | Mogadouro | Tó | 2021-07-20 | 2021-07-20 | 253 | 5 | 2021040019425 | 99 | 98 |

p1: stands for percentage of known fire progression (%); p2: stands for percentage fire behaviour descriptors calculated (%)



**Table A3. Attribute fields of the fire progressions (L1)**

| Field | Description | Possible values |
|---|---|---|
| id | Polygon ID | >0 |
| type | Type of Spread Polygon | p - wildfire progression ; z - ignition or active flaming zone ; a - previously burned area |
| date_hour | Date and hour of the polygon | yyyy-mm-dd hh:mm; uncertain ; na (not applicable) |
| source | Source of the data | fserv - forest service ; sat - satellite data ; airb - airborne data; fops - fire personnel; ek - expert knowledge; rep - external reports |
| zp_link | Numerical link between a ignition or active flaming zone ("z") polygon and a wildfire progression ("p") polygon | 1,2,3... - the link between types "p" and "z" with known dates and hours; 0 - used for type "a" or  when progression in "uncertain" or  when the link between "p" and "z" is unknown |
| burn_period | Burning period | 1,2,3,..; 0 for the same cases as "zp_link". |




**Table A4. Attribute fields of the fire behaviour database (L2)**

| Field | Description | Possible values |
|---|---|---|
| fid | Fire ID | 1-80* |
| fname | Fire Name | Municipality_StartDate (e.g. Gouveia_10082015) |
| year | Year | 2015-2021* |
| type | Type of Spread Polygon | p - wildfire progression ; z - ignition or active flaming zone ; a - previously burned area |
| sdate | Start date and hour of the polygon | yyyy-mm-dd hh:mm; uncertain ; na (not applicable) |
| edate | End date and hour of the polygon | yyyy-mm-dd hh:mm; uncertain ; na (not applicable) |
| inidoy | Start day-of-year of the polygon (hours in decimal values) | 1 to 366; -1 for uncertain progression polygons, polygons with unknown zp_link and previously burned areas |
| enddoy | End day-of-year of the polygon (hours in decimal values) | 1 to 366; -1 for uncertain progression polygons, polygons with unknown zp_link and previously burned areas |
| source | Source of the data | fserv - forest service ; sat - satellite data ; airb - airborne data; fops - fire personnel; ek - expert knowledge; rep - external reports |
| zp_link | Numerical link between a ignition or active flaming zone ("z") polygon and a wildfire progression ("p") polygon | 1,2,3... - the link between types "p" and "z" with known dates and hours; 0 - used for type "a" or when progression in "uncertain" or when the link between "p" and "z" is unknown |
| burn_period | Burning period | 1,2,3,..; 0 for the same cases as "zp_link". |
| area | Burned area extent (ha) | > 0 for progression polygons, -1 for ignition or active flaming zones. |
| growth_rate | Fire growth rate (ha/h) | >0 for progression polygons with zp_link value >0; -1 for previously burned areas or uncertain progression polygons |
| ros_i | Average rate-of-spread (m/h) calculated since ignition\active flaming areas or a progression marking the start of the burning period | >0 for progression polygons with zp_link value >0; -1 for previously burned areas or uncertain progression polygons |
| ros_p | Parcial rate-of-spread (m/h) calculated between consecutive ignition\active flaming areas and progression polygon, or between two consecutive progression polygons | >0 for progression polygons with zp_link value >0; -1 for previously burned areas or uncertain progression polygons |
| spdir_i | Spread direction associated with "ros_i" ( ° from North) | 0 to 359.99; -1 for the same cases in "ros_i" |
| spdir_p | Spread direction associated with "ros_p" ( ° from North) | 0 to 359.99; -1 for the same cases in "ros_p" |
| duration_i | Duration (hours) associated with the "ros_i" metric | >0 known progression polygons; -1 for ignition\active flaming zones, previously burned áreas or uncertain progression polygons |



| duration_p | Duration (hours) associated with the "ros_p" metric | >0 known progression polygons; -1 for ignition\active flaming zones, previously burned áreas or uncertain progression polygons |
|---|---|---|
| qc | Confidence flag for each wildfire | See table A1 |
| FRE | Fire Radiative Energy (TJ) | >0 for known progressions with at least 70% of FRE observations between "sdate" and "edate"; - 1 for the remaining polygons |
| FRE_flux | Fire Radiative Energy flux (TJ ha$^{-1}$ h$^{-1}$) | >0 for known progressions with at least 70% of FRE observations between "sdate" and "edate"; - 1 for the remaining polygons |
| FRE_perc | Percentage of FRE observations between "sdate" and "edate" | Between 0 and 100 for known progression polygons; -1 for the remaining. |

* values will change when the database will be updated with new wildfires.



**Table A5. Attribute fields of the simplified fire behaviour database (L3)**

| Field | Description | Possible values |
|---|---|---|
| fid | Fire ID | 1-80* |
| fname | Fire Name | Municipality_StartDate (e.g. Gouveia_10082015) |
| burn_period | Burning period | ⩾1 |
| year | Year | 2015-2021* |
| sdate | Start date and hour of the burning period | yyyy-mm-dd hh:mm; "na" for burning periods which only have progression polygons with unknown "zp_link" (see Table A4) |
| edate | End date and hour of the burning period | yyyy-mm-dd hh:mm; "na" for burning periods which only have progression polygons with unknown "zp_link" (see Table A4) |
| inidoy | Start day-of-year of the burning period (hours in decimal values) | 1 to 366; -1 for burning periods which only have progression polygons with unknown "zp_link" (see Table A4) |
| enddoy | End day-of-year of the burning period (hours in decimal values) | 1 to 366; -1 for burning periods which only have progression polygons with unknown "zp_link" (see Table A4) |
| qc | Confidence flag for each wildfire | See table A1 |
| area | Burned area extent (ha) | >0 |
| growth_rate | Average fire growth rate (ha/h) | >0; -1 for burning periods which only have progression polygons with unknown "zp_link" (see Table A4) |
| ros | Average rate-of-spread (m/h) | >0; -1 for burning periods which only have progression polygons with unknown "zp_link" (see Table A4) |
| max_ros | Maximum rate-of-spread (m/h) observed in the burning period | >0; -1 for burning periods which only have progression polygons with unknown "zp_link" (see Table A4) |
| spdir | Spread direction associated with "ros_i" ( ° from North) | 0 to 359.99; -1 for burning periods which only have progression polygons with unknown "zp_link" (see Table A4) |
| duration | Duration (hours) of the burning period | >0; -1 for burning periods which only have progression polygons with unknown "zp_link" (see Table A4) |
| FRE | Fire Radiative Energy (TJ) | >0 for known progressions with at least 70% of the area burned during the burning period covered with FRE estimates; - 1 for the remaining polygons |
| FRE_flux | Fire Radiative Energy flux (TJ ha$^{-1}$ h$^{-1}$) | >0 for known progressions with at least 70% of the area burned during the burning period covered with FRE estimates; - 1 for the remaining polygons |
| FRE_perc | Percentage of FRE observations between "sdate" and "edate" | Between 0 and 100 |

* values will change when the database will be updated with new wildfires.