# Peer review of "The Portuguese Large Wildfire Spread Database (PT-FireSprd)"

_Earth System Science Data, 2022_

## Author Comment (AC1)

**Reviewer #1**

Major Comments

1. The manuscript reads well, but there are a number of sentences that seem incomplete or are somewhat incoherent. I note those in the attached pdf. Authors should also be more direct and concise in their writing. The manuscript is quite long, and I made note of content that could be left out as it is not necessary.

We thank the reviewer for identifying the incomplete\incoherent sentences, and the content that could be left out. All comments that were annotated in the PDF are replied below under "minor comments", unless the comment itself is already addressed in the "major comments". We reviewed the manuscript and tried to implement a more direct and concise writing.

2. This is a wildfire spread database which is quite relevant. The authors should nonetheless mention clearly that no collation of data on weather, fuels, topography, etc was conducted within the context of this study. This is never mentioned. Although the authors mentioned later in the manuscript that the data can be used for better understanding of wildfire drivers, model evaluation, etc. this cannot be conducted unless other data is present. As it is, the spread data in isolation does not allow for much of an analysis.

This information is given to the reader very clearly at the end of the Introduction: "Fire behaviour is described sensu stricto, thus analysis of its drivers and effects is beyond the scope of the current work.".

We followed the reviewer's suggestion and specifically stated that the fire behavior data must be combined with detailed driver/effect data to perform the potential analysis suggested in the Discussion section.

3. There are important limitations of the satellite data. Some of them are mentioned, others are not. An important point that should be made is that just because the satellite or someone says the fire is at a certain location at a given time, it does not mean that the fire just arrived there at such time. The fire might have arrived hours before, and hence the average rate of spread for a burning period is a value that is diluted, combining periods of rapid spread and no spread. This is quite relevant as fire behaviour is highly nonlinear, and averaged values over larger time periods can be misleading. These aspects should be noted in the manuscript.

First, an overview of satellite data limitations and uncertainties is already provided in the 7[th] paragraph of the Introduction. In the Discussion section, we specifically state that the limitations discussed there are specific for Portugal, since the more generic limitations have been previously described.

Second, the description of how uncertainties regarding "progression polygon date\time" were already partially written in section 2.3, but we acknowledge and agree with the reviewer that additional clarification is needed. Therefore, we added a justification in section 2.3 that now reads: "A common challenge found in the delineation of the wildfire progression were the uncertainties associated with the correct time an entire progression polygon burned. These uncertainties were present in almost all data sources. For example, a polygon derived by fire

operatives on the ground could have stopped burning minutes or hours before data collection. Additionally, satellite active-fire data can depict areas that are hot minutes or hours after the fire front stopped progressing. The strategy to minimize such uncertainties was to use data from multiple sources, seeking convergence of evidence."

This text was inserted before the paragraph that was already written in the first version of the manuscript which provides an example of how multi-source data were used to reduce such uncertainties. We would like to highlight the last sentence of that paragraph: "When data were insufficient to determine when a given area burned, the spread polygon was flagged as "uncertain"."

4. It is not clear how combining different methods to map a fire location are integrated. Do some methods have prevalence over others? Photographic evidence vs satellite information? Is there a process that is followed? If yes, this should be described.

We agree with the reviewer that this information was missing. We have added it to the Methods (section 2.3) that now reads: "The progression polygons were built using as many data sources as possible, complementing each other in both space and time (see Figure 1). The variety of input data used have different associated uncertainties. When delineating the progression polygons priority was given to input data with higher spatial resolution, free from smoke and cloud contamination, and with the most complete view of the entire active part of the wildfire. Typically, the first priority level data (i.e. highest confidence) were Sentinel-2 and Landsat 8/9 images, and AVRAC aeroplane photos/videos. The second priority level was composed of ground data, VIIRS active-fires, PROBA-V and Sentinel 3 images (both at 300 m resolution) and helicopter photos/videos. The third priority level were images and active-fire data from moderate resolution satellites (MODIS and Sentinel 3). The fourth, and last priority level (i.e. lowest confidence) were composed by FRP data from MSG-SEVIRI and the official wildfire time logs. The data from the large 2017 wildfires reports were handled separately. The progression polygons from Guerreiro et al., (2017, 2018) were deemed as high confidence data and were complemented with data and information from Viegas et al. (2019) and, at times, with satellite data."

5. I have two main technical comments.

   1. The proposal of a fire behaviour classes based on your data is fraught with error. It is ok to explore the distribution of your data, but to propose such distribution (which you said, was biased to large fires) to derive a fire behaviour classification is wrong. The classification class threshold have no physical meaning, and you can realise that if you check into a number of fire behaviour and danger classifications developed from fire behaviour – operational implications. A proof that your proposed classification is meaningless, is the fact that if in the next two fire seasons you add 40 new wildfires all burning under moderate to high fire danger (lets say it is a mild fire season), your new fire behaviour classification classes will change drastically. What is the point then? I strongly suggest this is removed from the manuscript.

We partially agree with the reviewer in the sense that "fire behavior classes" already exist, have physical and operational meaning and using the same term in our work would not be correct. Therefore, we propose to replace "fire behavior classes" by "fire behavior percentiles". We do not propose (anywhere) that these percentiles are related to type of fire and intensity.

The distributions are meaningful for several reasons. Fire behavior classes are very useful, but do not add information regarding the frequency each class has been observed. However, the latter is possible by using a statistical approach to describe the distribution of observed fire behavior. Distributions are meaningful to perform analyses, e.g.: fire behavior variability within a wildfire, between several wildfires of different countries/regions/fuel types, between several wildfires in different years or within the same year. In the long run, these distributions can even support better fire regime characterization, which has been added to the Discussion as potential application of the database.

The addition of new observations to a distribution does not mean it the original distribution was irrelevant, on the contrary, it contributes to make it more representative. Following the example given by the reviewer, if in the next two fire seasons we add 40 wildfires burning under moderate conditions, the distribution of fire behavior will be even more representative of the entire "population" (i.e. behavior of large wildfires). The extreme values will most likely continue to be mostly associated with the progression polygons of June and October 2017 wildfires. This is statistically coherent since extreme values are by definition the ones located in the tails of the distribution, in this specific case, the right tail. The large impact stated by the reviewer is unlikely to occur, since the original distribution presented in this paper has around 900 observations of ROS and FGR, and can be considered statistically robust.

If 2017 had never happened, the distribution would have been different, and so would the percentiles. With the occurrence of wildfires similar to those observed in 2017 the distributions would have changed and the higher percentile intervals would have been redefined. This was exactly what happened in the hot summer of 2010 and led to a redrawing of the temperature record maps in Europe, as presented by Barriopedro et al. (2011), published in Science.

As detailed in the Discussion section, and also mentioned by the reviewer, the distribution of the fire behavior descriptors is biased towards larger wildfires. We acknowledge in the manuscript (Discussion section) the need to include add smaller wildfires. However, it is not certain that including smaller wildfires (even if larger than 100ha) will significantly change the distribution of the fire behavior descriptors. From our experience and also based on the database analysis, smaller wildfires do not necessarily mean low ROS values. For example, relatively small but fast spread cropland fires, or wildfires that spread towards a fuel discontinuity\barrier. At the other end, some medium\large wildfires have small ROS values because they burn in locations where suppression is very limited (e.g. in areas with low road accessibility where only aerial means operate). The inclusion of new wildfires will change the percentiles, like it would in any other distribution, but this impact needs to be evaluated with time.

The fire behavior descriptors were calculated for several wildfires, and for different parts of the same wildfire, thus encompassing a large variability and range of values. The percentiles are useful to easily communicate the distribution of the fire behavior descriptors, which has its value both for researchers and for practitioners. The approach is merely statistical. The distributions have a clear spatial (Portugal) and temporal (2015-2021) context, focusing mainly on large and very large wildfires. The distributions are relevant for researchers, providing them with useful statistical information regarding the fire behavior that goes beyond the mere final burned extent, and to better frame observations with existing fire behavior classes (e.g. Hirsch & Martell 1996, Alexander and Lannovile, 1990 or Tedim et al. 2018). The distributions are also relevant for practitioners (who, by the way, contributed to this study) providing them with concrete references of fire behavior for their country relatively to those from other countries with distinct contexts. It will also allow them to better analyze historical wildfires and to frame new ones (not

contained in PT-FIRESprd) by comparing them with historical fire behavior. For example: how did the fire behavior of wildfire "X" in 2022 compare with historical data? (this has been recently done together with fire operatives).

2. The authors use their dataset and make a 'finding' that area burned is mostly a function of fire growth rate rather than rate of fire spread. This result is obvious by several reasons, the simplest one being that the rate of spread is only related to the area burned for the initial stages of a fire growing from a point source. From the moment a fire is affected by topography, fuels, and burn over several burn periods and days, it is the area growth rate that is linked with the fire area, not the rate of fire spread. I do not see this a finding, whatsoever. Of course, a 2 dimensional area growth metric is going to be more related to the final burned area than a one dimensional metric of fire propagation (ROS). As with the previous point, I strongly suggest that this is removed from the manuscript.

We understand the reviewer's point, but we disagree for several reasons. The relationship between ROS, FGR and burned area extent depends mostly on the geometrical format of the fire progression. Elongated fire perimeters (typically driven by strong unidirectional winds) will likely have large ROS but will not necessarily lead to very large burned area extent. Conversely, less elongated wildfires may lead to large burned extents under moderate ROS. We are not aware of any study that explains the factors behind these differences. If the reviewer is aware of past studies that analyze the relation between these three descriptors, we kindly ask him to provide us with the references.

The fact that the burned area extent is more related with FGR than the ROS is relevant for various applications. Some researchers and fire analysts, use ROS as the most important descriptor to predict\evaluate the potential behavior of a wildfire. What our analysis suggests is that, by doing so, they may be only partially evaluating the fire potential, because burned area extent will only be moderately correlated with ROS. By integrating in their analysis, FGR predictions\evaluations, their analysis will likely to provide more solid results regarding the potential burned extent of a specific wildfire. Burned extent is an important descriptor for a fire managers and incident commanders. These issues are typical in wildfire spread simulations where models are calibrated\evaluated considering ROS and have poor ability to estimate burned area extent (in our experience, a good estimation of ROS overpredicts fire size and fire growth rate). It is beyond the scope of this review, and the report itself, to understand the main drivers behind such discrepancies. However, this does raise relevant questions such as: what are the drivers of FGR and how to these differ from the well-studied ROS drivers?

Finally, we can provide two practical examples contained in the PT-FIRESprd. The Pedrogão Grande 2017 wildfire had his largest ROS and FGR when a lengthy flank transformed into the head of the fire. Before that moment the ROS was high, but the FGR was low (wind driven fire), and afterwards both ROS and FGR were very high. In the Monchique 2021 wildfire, an extensive flank also became the head of the fire, however with a very low ROS and proportionally large FGR. Operationally this information is relevant because the fact that ROS was low did not mean that there weren't significant operational challenges to suppress the fire given its large FGR.

It is not clear why the authors depart from their main focus of the study, describing how the database was assemble, to do a spurious analysis of the data and come up with these findings, that, in my view, are not really findings. If the authors want to explore those aspects of fire behaviour, then they should do so in a different piece of work, with proper basis and analysis.

The objective of the work was to develop a fire spread database focusing on fire behavior descriptors in a "strict sense", i.e. without considering drivers or effects. The two main aspects of analysis that the reviewer refers to, were motivated by the need to show to the reader (1) the distribution of the fire behavior descriptors and (2) highlight some basic relationships between them. It is beyond the scope of the work to understand the drivers. It would also make the work less understandable by the readers if we explained them how we did it, without exploring the database (in strict sense).

**Minor comments**

- L29: Changed precision

- L29: Rephrased

- L44: "Propagation mode" can be, for example, surface or crown fire. We removed this term, and the term "perimeter" to avoid confusion, and we have already enough examples of common metrics in that sentence

- L44: Regarding the remaining text highlighted, we do not understand what the reviewer wants to point out, since these are common fire behavior descriptors.

- L49-50: rephrased the sentence.

- Regarding the comment regarding the incorrect/excessive use of the reference Gollner et al. 2015 *"5 citations of this report in two paragraphs. Comes across as it being the sole source of your understanding - and that you base much of your intro in a sole source. i would suggest you reference only when strictly necessary"*. We disagree with the reviewer's comment for the following reasons:

> 1. First and foremost, we have used around 40 different references in the Introduction, so Gollner et al. (2015) is hardly our sole source of information.
>
> 2. Second, the work of Gollner et al. (2015) contains an overview of fire behavior modelling capabilities, limitations and improvements foreseen in the near future. The report summarizes the contributions of several key experts done in a workshop. Therefore, it contains much information that is relevant for the entire work with the advantage of containing several different aspects of fire behavior modeling in a single document.
>
> 3. Gollner et al. 2015 was used mostly as reference for the following topics\messages: i) the importance of detailed open fire behavior data; ii) mapping fire front progression under different environmental conditions; iii) the limitations that undermine the availability of good quality open-access data on fire behavior; iv) the requirements for fire progression observations.

We agree that the reference in line 67 was incorrectly assigned and therefore removed it.

- L81: We agree with the reviewer's suggestion of adding "wildfire propagation", however, we prefer not to include the importance of remote sensing in providing relevant information regarding the drivers (e.g. fuel, Marta Yebra's work is a good example), nor impacts, since the report is focused on fire spread in a "strict sense".

- L83: ""this is upward only right? radiation is emitted in all directions and this radiative power is a course remote sensed metric" FRP is the rate of energy released associated with a particular pixel identified by an active fire. It is based on the atmospherically corrected radiance difference, between the fire pixel and its surrounding pixels, weighted by the pixel area. These radiances are captured by the satellite sensor in the MWIR during its scanning swath. Assuming no atmospheric effects, a pixel at nadir would be dominated by the radiance emitted at 90º, and a pixel at the edge of the swath would capture the radiance at lower angle. The radiation model is considered near-lambertian so the direction variability is not an issue. The resolution depends on the sensor, VIIRS as a 375m spatial resolution which is cannot be considered coarse, particularly considering the extent of the wildfires in the database. On the other hand, MSG-SEVIRI provides very coarse estimates of FRP (~4km at Portugal's latitude).

Coen & Riggan (2014) used radiometric temperature temperature to assess estimated fireline intensity. Regardless, the reference is incorrect as they did not use FRP, neither did the other authors referenced in L83, therefore we removed that part of the sentence.

-L98: We added two additional references: Storey et al. 2021 and Stow et al. 2014

-L125: added reviewers suggestion

-L132: added information about the total number of wildfires, and the number of wildfires > 100ha.

-L157-161: we agree and have deleted the entire paragraph from the Methods section and integrated parts of it in the Introduction (5[th] paragraph)

- L175: we added "atmospherically corrected (Level 2)"

- L221: Changed "forest service" to ICNF

- L225: Added resolution of Copernicus Emergency Management Service data

- L226: We followed the reviewer's suggestions.

- L231: We understand this can be confusing. The ignition time is provided by the SADO system operated by the civil protection (ANEPC). The official final ignition location is provided by ICNF, most of the times after post-fire investigation, however, the SADO system contains the estimated ignition location provided by first responders. Both are rarely equal, and complement each other when trying to reconstruct the first step of the fire progression. Regarding the "time log": we cannot remove it, since it was one of the pieces of information that we used to reconstruct fire spread. Summarizing: We merged parts of the text and reduced its length.

- L234: Added "Reports of the 2017 large wildfires"

- L252: We have rephrased the entire paragraph that now reads: "Persistent cloud cover hindered the June and October 2017 wildfires progression mapping with satellite data. Nonetheless, given the relevance of these wildfires we decided to include these fire progressions in our database because they represent some of the largest and most extreme wildfires that ever occurred in mainland Portugal."

- L268: For two reasons: 1) To guarantee that the fire perimeter does not contain areas burned in different wildfires (we found some cases where this error was present in the official fire database); 2) For fire spread simulation studies, enabling them to mask "previously burned" areas (ignoring these areas can have a relevant negative impact on the simulation)

- L364: Changed the minute symbol in this line and in the rest of the document

- L373: Changed "direction of forward spread". The uncertainties, as discussed in the document, vary with time and data source. We believe changing the units from m/h to km/h will have little impact on the reader's perception of uncertainty.

- L489: Addressed in "major comments"

- L500: Changed to "shows"

- L550: We added that information to the start of the sentence, that now reads: "Combined with detailed information on the drivers, namely weather and fuel, and its effects, it can be used to"

- L555: Addressed in "major comments"

- L559: The answer to this question is provided 2 sentences below.

- L563: It is a "may" because it needs to be shown that including smaller wildfires will significantly change the distribution of the fire behavior descriptors. At this point this is not clear. From our experience and also based on the database analysis, including smaller wildfires will not necessarily mean that we will include smaller ROS values (e.g. relatively small but fast spread cropland fires). At the other end, some large wildfires have small ROS values because they burn in locations were suppression is impossible or very difficult.

- L566: Replied in "major comments"

- L587: one value is for the "progression polygons" and the other is for the "burning periods". We have added L2 and L3 to make the distinction clearer.

- L609: replied in "major comments"

- L621 and 622: We understand that aircraft is more encompassing, but some level of distinction needs to be made here. Initial attack photos are taken by operative personnel onboard helicopters, not aeroplanes. The second sentence should be more generic because both helicopters and aeroplanes are included, and therefore we changed "aeroplane" to "aircraft"

- L637: "throughout the methods there is no descriptino of the resolution of the satelite data used, and its implications in data uncertainty."

We do not understand this comment, considering that throughout the entire Methods section, the spatial resolution of all satellite data used are presented. Based on a suggestion of another reviewer we have added in the revised version a table in the Supplements listing the characteristics of all input data used.

Regarding uncertainty, the paragraph added to section 2.3 as a response to one of the reviewer's major comments, addresses uncertainty "When delineating the progression polygons priority was given to input data with higher resolution, smoke and cloud free, and with the most complete view of the entire active part of the wildfire.". Thus, lower resolution satellite data was considered as more uncertain than higher resolution data.

- L657: A reference of FRM will be good. We added a reference: Niro et al. 2021.

**Reviewer #2**

Benali et al. have developed a valuable wildfire behaviour dataset that includes the spread of 80 large wildfires in Portugal between 2015 and 2021. The authors have combined data from multiple sources, which helps to reduce gaps and uncertainties in the wildfire data collection. This open data has the potential to improve the simulation of wildfire in the context of changing environments and to better manage wildfires. The manuscript is well-written and the authors have done a commendable job of presenting the data. I think it is publishable if several minor issues can be addressed.

The Input Data section could benefit from being more concise. Consider presenting the information in tables, especially when discussing data sources, to improve clarity for readers.

We revised the Input data section to make it shorter. We added a table in the Supplements (Table A1 of the revised manuscript) with a list of the input data used and its main characteristics.

In section 3.1 Overview of the PT-FireSprd database, the authors provide a comparison of ROS and FGR and give some preliminary results. However, this appears to detract from the main focus of the paper, and as a result, the section feels disjointed. Consider revising the section to more clearly tie to the central argument of the paper or move the comparison to an appendix to maintain focus on the main topic.

The main focus of the work was the development of the fire spread database. We believe it is very important to provide credibility to the reader by showing the database from different perspectives, that will possibly motivate different applications. Therefore, we showed maps of ROS for some wildfires, several fire behavior descriptors for one wildfire but detailed over time and finally the distribution of main fire behavior variables and how they are related. We believe moving outside of the main topic would be, for example, to analyze the drivers of fire behavior.

This approach is aligned with previous publications of ESSD regarding wildfires, for example, Andela et al 2019 (doi:10.5194/essd-11-529-2019) that illustrates the use of their database to characterize fire regimes (section 3.2) and identify fire extremes (section 3.3)

**Minor comments:**

Line 508: It should be 3.2 not 2.2. Changed accordingly.

**Reviewer #3**

This manuscript details the creation of the first truly multi-proxy high spatial and temporal resolution fire progression archive of its kind. Combining data from multiple remote sensing sources leveraging the full range of available temporal and spatial resolution, combined with empirical observations, written records, and photographic evidence, this is the most complete crosswalk of multi-proxy data sources I have seen. The dataset has significant potential to improve the understanding of fire activity, resource use effectiveness, fire climatology, and other fields in Portugal.

The data are catalogued in three phases, the first in my option being the largest contribution and most important- a structured approach to reconstructing fire progression at the finest resolution possible. The second phase develops three derived descriptor variables that include some straightforward calculations under the stated assumptions (rate of spread and rate of fire growth) and some more questionable calculations (average fire radiative power) given the nature of the data sources. The third phase leverages the derived characteristics to ascribe a fire behavior class to each fire growth polygon. I'm not clear on how thresholds were determined to differentiate fire behavior classes or how they would be used operationally but if determined in consultation with fire managers I can see this as a useful way to present the data.

Acknowledging that "fire behavior classes" already exist (e.g. Tedim et al. 2018; Alexander and Lannoville,1990) and have physical and operational meaning, using the same term in our work could lead to a misinterpretation. Therefore, we replaced "fire behavior classes" simply by "fire behavior percentiles".

Fire behavior classes are very useful, but do not add information regarding the frequency each class has been observed. However, the latter is possible by using a statistical approach to describe the distribution of observed fire behavior. Distributions are meaningful to perform analysis, e.g.: fire behavior variability within a wildfire, between several wildfires of different countries/regions/vegetation types, between several wildfires in different years or within the same year. On the long run, these distributions can even support better fire regime characterization, which has been added to the Discussion as potential application of the database.

The fire behavior descriptors were calculated for several wildfires, and for different parts of the same wildfire, thus encompassing a large variability and range of values. The percentiles are useful to easily communicate the distribution of the fire behavior descriptors, which has its value both for researchers and for practitioners. The approach is merely statistical. The distributions have a clear spatial (Portugal) and temporal (2015-2021) context, focusing mainly on large and very large wildfires. The distributions are relevant for researchers, providing them with useful statistical information regarding the fire behavior that goes beyond the mere final burned extent, and to better frame observations with existing fire behavior classes (e.g. Hirsch & Martell 1996, Alexander and Lannovile, 1990 or Tedim et al. 2018). The distributions are also relevant for practitioners (which can be easily verified by their contribution to this work) providing them with concrete references of fire behavior for their country opposed to the ones regarding other countries with distinct contexts. It will also allow them to better analyze historical wildfires and to frame new ones (not contained in PT-FIRESprd) by comparing them with historical fire

behavior. For example: how did the fire behavior of wildfire "X" in 2022 compare with historical data? (this has been done in practice recently, by fire operatives).

**Major concerns:**

The treatment of Fire Radiative Energy as an additive measurement or something that can divided by an area is problematic. By definition, fire radiative energy is an instantaneous and constantly varying measure of energy release for a given area of measurement (Zhang et al 2018). Dividing FRE by an additive area metric to divvy the instantaneous measure by the area burned assumes that the area burned occurred in that instant. The area burned is a function of the free burning rate per 30-minute period. This is mixing average and instantaneous data sources. Unless the MSG_SEVIRI sensor does this different from MODIS, I don't think it can be applied to an area polygon.

FRE, measured in Joules, is the integrated estimation of the energy that was released by a fire, and it is not an instantaneous measure. FRP, measured in Watts, is the instantaneous measure from which FRE can be estimated. In this study, the sum of all pixels for which MSG-SEVIRI FRP estimates were associated with an active fire, represent its total rate of energy power. To estimate FRE, we have assumed that this rate remains constant during the very short detection frequency (15 min). Therefore, FRE is an extensive physical quantity, in time and space, that can be divided by the area to provide an indication of energy release rate, or also known by fluence. For example, since FRE is directly proportional to the dry matter consumed it is used to estimate biomass burning emissions. We acknowledge that this was not completely clear and revised the methods at the end of section 2.2.1. We also added a reference (Pinto et al. 2018) that estimated FRE using the same approach. Thus, FRE is not instantaneous and reflects the sum of energy released by the wildfire every 15min (i.e. an integration).

The main challenges of using the MSG-SEVIRI FRP product is its coarse resolution, which hinders the ability to identify accurately to which progression it belongs. We have minimized this by using an association algorithm that takes in to account space and time. These assumptions and limitations are described in the text. Considering that we know the total area burned during a specific time interval, we can associate the energy released for the same period as being associated with that burned extent. The process has uncertainties: we made a simplification by considering that, in the same time period, larger burning polygons contributed more (and linearly) to the total FRE, than smaller polygons. We hope that this explanation clarifies the doubts raised by the reviewer.

**Minor concerns:**

The manual methods are quite labor intensive and leave room for standardization and automation that would alleviate some concerns I have about repeatability with other data sources and in other geographies.

The potential future improvements have been stated in the Discussion section, particularly the development of automated methods to delimit fire progression using airborne and satellite data, as well as, regarding the overall fire progression methodology.

**Other comments:**

I agree that and change detection algorithm applied to the 30-minute FRE, ROS, or FRG could be an interesting way to determine the important phases of fire growth and then relate these phases to other environmental data (e.g. weather, fuel matrix, suppression resources use, etc.). I hope the research group is able to continue updating the dataset and is able to adapt systems for standardizing quality control of records and automating generation of polygons and derived metrics. As it is the dataset serves as a valuable series of inputs for future analysis and keeping it up to date will ensure continued use and applications.

We thank the reviewer for the comment and suggestions.